# SKIP-HOPS recruits TBC1D15 for a Rab7-to-Arl8b identity switch to control late endosome transport

Marlieke LM Jongsma[1,2,†], Jeroen Bakker[1,2,†], Birol Cabukusta[1,2,‡], Nalan Liv[3,‡], Daphne van Elsland[1,2], Job Fermie[3] , Jimmy LL Akkermans[1,2], Coenraad Kuijl[4] , Sabina Y van der Zanden[1,2], Lennert Janssen[1,2], Denise Hoogzaad[1], Rik van der Kant[5], Ruud H Wijdeven[1,2], Judith Klumperman[3], Ilana Berlin[1,2,*] & Jacques Neefjes[1,2,**]

## Abstract

The endolysosomal system fulfils a myriad of cellular functions predicated on regulated membrane identity progressions, collectively termed maturation. Mature or "late" endosomes are designated by small membrane-bound GTPases Rab7 and Arl8b, which can either operate independently or collaborate to form a joint compartment. Whether, and how, Rab7 and Arl8b resolve this hybrid identity compartment to regain functional autonomy is unknown. Here, we report that Arl8b employs its effector SKIP to instigate inactivation and removal of Rab7 from select membranes. We find that SKIP interacts with Rab7 and functions as its negative effector, delivering the cognate GAP, TBC1D15. Recruitment of TBC1D15 to SKIP occurs via the HOPS complex, whose assembly is facilitated by contacts between Rab7 and the KMI motif of SKIP. Consequently, SKIP mediates reinstatement of single identity Arl8b sub-compartment through an ordered Rab7-to-Arl8b handover, and, together with Rab7's positive effector RILP, enforces spatial, temporal and morphological compartmentalization of endolysosomal organelles.

**Keywords** Arl8b; HOPS; Rab7; SKIP; TBC1D15
**Subject Category** Membranes & Trafficking
The EMBO Journal (2020) 39: e102301

## Introduction

The endolysosomal system consists of a dynamic network of vesicular structures working together to achieve controlled uptake and proteolysis of materials derived from the extracellular space.

Regulated transport and sorting of cargoes along the endocytic route enable cells to effectively interpret and mitigate extracellular cues (Sigismund *et al*, 2012; Bakker *et al*, 2017), fight off pathogens (Gruenberg & van der Goot, 2006; Roche & Furuta, 2015) and sustain homeostasis (Lim & Zoncu, 2016). To ensure order within this vesicular network and regulate access to the proteolytic compartment, the endolysosomal system is compartmentalized in cellular space (Neefjes *et al*, 2017). Once nascent endosomes are acquired in the cell periphery, they undergo progressive maturation through controlled interactions with late compartments (Huotari & Helenius, 2011). This process of maturation is accompanied by ordered transitions in membrane identity (Cullen & Carlton, 2012), coupled to sequestration of cargoes marked for destruction onto intraluminal vesicles (ILVs; Christ *et al*, 2017). The resulting multi-vesicular bodies (MVBs; Woodman & Futter, 2008) can subsequently fuse with lysosomes to deliver their luminal contents for degradation (Klumperman & Raposo, 2014; Huber & Teis, 2016). In addition to proteolysis, MVBs and lysosomes carry out diverse cellular functions, including nutrient sensing (Sancak *et al*, 2007, 2008; Korolchuk *et al*, 2011), processing and loading of antigens for presentation (Kleijmeer *et al*, 2001) and exosome secretion for long-range communication between cells (Raposo & Stoorvogel, 2013). It is becoming increasingly clear that efficient and timely execution of their functions is intimately linked to the intracellular location and motility of late organelles (Neefjes *et al*, 2017); however, ways in which such attributes are regulated in space and time remain ambiguous. Here, we explore one aspect of this complexity by dissecting how changes in endolysosomal membrane identity are coupled to choice(s) of transport route.

To ensure productive interactions between diverse vesicles within the network and licence appropriate cargo flow, endosome identity must be clearly defined and easily interpreted. This identity, or maturation status, relies on progressive changes in the

1 Department of Cell and Chemical Biology, Leiden University Medical Center (LUMC), Leiden, The Netherlands
2 Department of Cell and Chemical Biology, Oncode Institute, Leiden University Medical Center, Leiden, The Netherlands
3 Section Cell Biology, Center for Molecular Medicine, University Medical Center Utrecht, Utrecht, The Netherlands
4 Department of Medical Microbiology and Infection Control, VU University Medical Center, Amsterdam,The Netherlands
5 Center for Neurogenomics and Cognitive Research, Faculty of Sciences, VU Amsterdam, Amsterdam, The Netherlands
*Corresponding author. Tel: +31 71 526 8729; E-mail: i.berlin@lumc.nl
**Corresponding author. Tel: +31 71 526 8727; E-mail: j.j.c.neefjes@lumc.nl
†These authors contributed equally to this work as first authors
‡These authors contributed equally to this work

endosome's phospholipid repertoire (Schink *et al*, 2016), coupled to membrane occupancy by small GTPases that constitute the principal drivers of vesicular traffic (Wandinger-Ness & Zerial, 2014). Molecular switches of this type alternate between a GTP-bound state, corresponding to the active membrane-bound form, and a GDP-bound inactive state. Hence, their membrane residence time and biological activity are modulated by cognate guanine exchange factors (GEFs) and GTPase-activating proteins (GAPs; Bos *et al*, 2007). For instance, while early endosomes typically carry the GTPase Rab5, late endosomes are instead marked by Rab7 (Zerial & McBride, 2001). During maturation, endosomes undergo a Rab5-to-Rab7 handover through an elegant mechanism, wherein Rab5 stimulates acquisition of Rab7 by recruiting its GEF, the Mon1/Ccz protein complex, while Rab7 in turn brings in the GAP for Rab5 (Nordmann *et al*, 2010; Poteryaev *et al*, 2010). This ordered transition serves as a gateway to the proteolytic compartment, establishing Rab7 as the central manager of transport and fusion events necessary for controlled delivery of endocytic, phagocytic and autophagic cargoes for degradation (Langemeyer *et al*, 2018). While, like Rab5, Rab7 can be removed from membranes through GAP-induced inactivation (Seaman *et al*, 2009; Carroll *et al*, 2013; Wong *et al*, 2018), whether it can also initiate a regulated handover to another GTPase is unknown.

Besides Rab7, late endosomes and lysosomes can also harbour Arl8b (Hofmann & Munro, 2006)—a GTPase implicated in diverse processes, including lysosomal degradation (Marwaha *et al*, 2017; Oka *et al*, 2017), antigen presentation and microbial killing (Garg *et al*, 2011), as well as nutrient sensing and autophagy (Jia *et al*, 2017; Pu *et al*, 2017). Arl8b is activated onto membranes by BORC, a multiprotein complex under the control of the mTOR pathway (Pu *et al*, 2015), and appears to coexist with Rab7 on a subset of late compartment structures (Mrakovic *et al*, 2012; Bento *et al*, 2013; Marwaha *et al*, 2017). By contrast to Rab7, which promotes perinuclear accumulation of the late compartment, acquisition of Arl8b is associated with positioning of vesicles in the periphery of the cell (Bonifacino & Neefjes, 2017). Due to the polar arrangement of microtubules, radiating from their juxtanuclear organizing centre (minus-end) towards the cell periphery (plus-end), movement into opposing directions involves at least two motor proteins—the dynein–dynactin complex for minus-end-directed transport, and one or more members of the kinesin family for transport to the plus-end (Granger *et al*, 2014). Desired motor complexes can be recruited to endosomal membranes on demand through appropriate GTPase effectors. Notably, Rab7 utilizes its effector RILP to recruit the dynein motor complex (Jordens *et al*, 2001), as well as associate with the HOmotypic fusion and Protein Sorting (HOPS) complex necessary for lysosomal degradation of materials contained within the MVB (van der Kant *et al*, 2015; McEwan *et al*, 2015a). On the other hand, Arl8b utilizes its effector SKIP/PLEKHM2 to recruit the kinesin-1 motor for transport towards the microtubule plus-end (Rosa-Ferreira & Munro, 2011). In addition, Arl8b can also attract the HOPS complex for fusion, either through SKIP (Khatter *et al*, 2015) or its family member PLEKHM1 (Marwaha *et al*, 2017). Recently, it was described that fusion between Rab7 and Arl8b vesicles gives rise to a hybrid identity compartment (Marwaha *et al*, 2017). However, what tactics the resulting endolysosomes employ to negotiate the choice between Rab7- versus Arl8b-directed transport routes remains unexplored. One option for ensuring spatiotemporal control under such

conditions would be to simply remove one of the two GTPases, resulting in an autonomous single identity compartment.

To interrogate the above possibility, we studied ways in which established transport effectors of Rab7 and Arl8b influence the identity and location of late endosomes and lysosomes. We find that, while Rab7 and Arl8b can reside on the same membranes, expression of opposing transport route effectors RILP and SKIP spatially segregates the endolysosomal repertoire between morphologically distinct perinuclear and peripheral pools, respectively, marked by Rab7/RILP and Arl8b/SKIP. This segregation is predicated on an ordered GTPase switch from Rab7 to Arl8b, occurring on membranes destined for Arl8b/SKIP-mediated transport to the cell periphery. The switch involves recruitment of the GAP TBC1D15 to SKIP via its associated HOPS complex. Then, the GAP becomes competent to inactivate and remove Rab7 from SKIP-positive membranes—a situation analogous to Rab7-induced removal of Rab5 from hybrid Rab5/Rab7 endosomes. The greater endolysosomal system is thus controlled by consecutive handover mechanisms, with Rab5-to-Rab7 transition followed by a Rab7-to-Arl8b switch for regulated membrane transport.

## Results

### Rab7 modulates the organization and dynamics of the Arl8b compartment

To study transport route selectivity operational on endolysosomes, we examined the interplay between their associated GTPases, Rab7, and Arl8b. In agreement with previous findings (Marwaha *et al*, 2017), we noted that part of the repertoire populated by Rab7 also carries Arl8b (Fig 1A and B). It has been speculated that Arl8b marks a later maturation stage as compared to Rab7 (Hofmann & Munro, 2006; Garg *et al*, 2011), and we therefore considered whether Rab7 activity status affects localization and behaviour of Arl8b. In cells co-expressing wild type mCherry-Rab7, vesicular repertoire carrying Arl8b-GFP displayed bilateral organization in space and time (Fig 1C and D, and Movie EV1) characteristic of late compartments (Jongsma *et al*, 2016), with the bulk of vesicles congregated in the perinuclear (PN) region exhibiting less movement as compared to their peripheral (PP) counterparts. Expression of a constitutively active Rab7 mutant Q67L, which, due to its inability to complete the GTP hydrolysis cycle, cannot be released from membranes (Mukhopadhyay *et al*, 1997), resulted in loss of this PN/PP distinction, yielding disorganized movement of Arl8b-positive vesicles throughout the cell (Fig 1C and D). By contrast, expression of a dominant negative Rab7 mutant T22N, harbouring an inactive GTPase domain (Mukhopadhyay *et al*, 1997), rendered the cell periphery largely devoid of Arl8b-positive endolysosomes, causing a marked decrease in their motility throughout the cell (Fig 1C–F, and Movies EV2 and EV3). These results imply that Rab7 activity status impacts the localization and behaviour of organelles targeted by Arl8b.

### Rab7/RILP and Arl8b/SKIP complexes mediate spatial segregation of late endosomes

To investigate the manner in which Arl8b and Rab7 coordinate spatial organization and motility of endolysosomes, we selectively

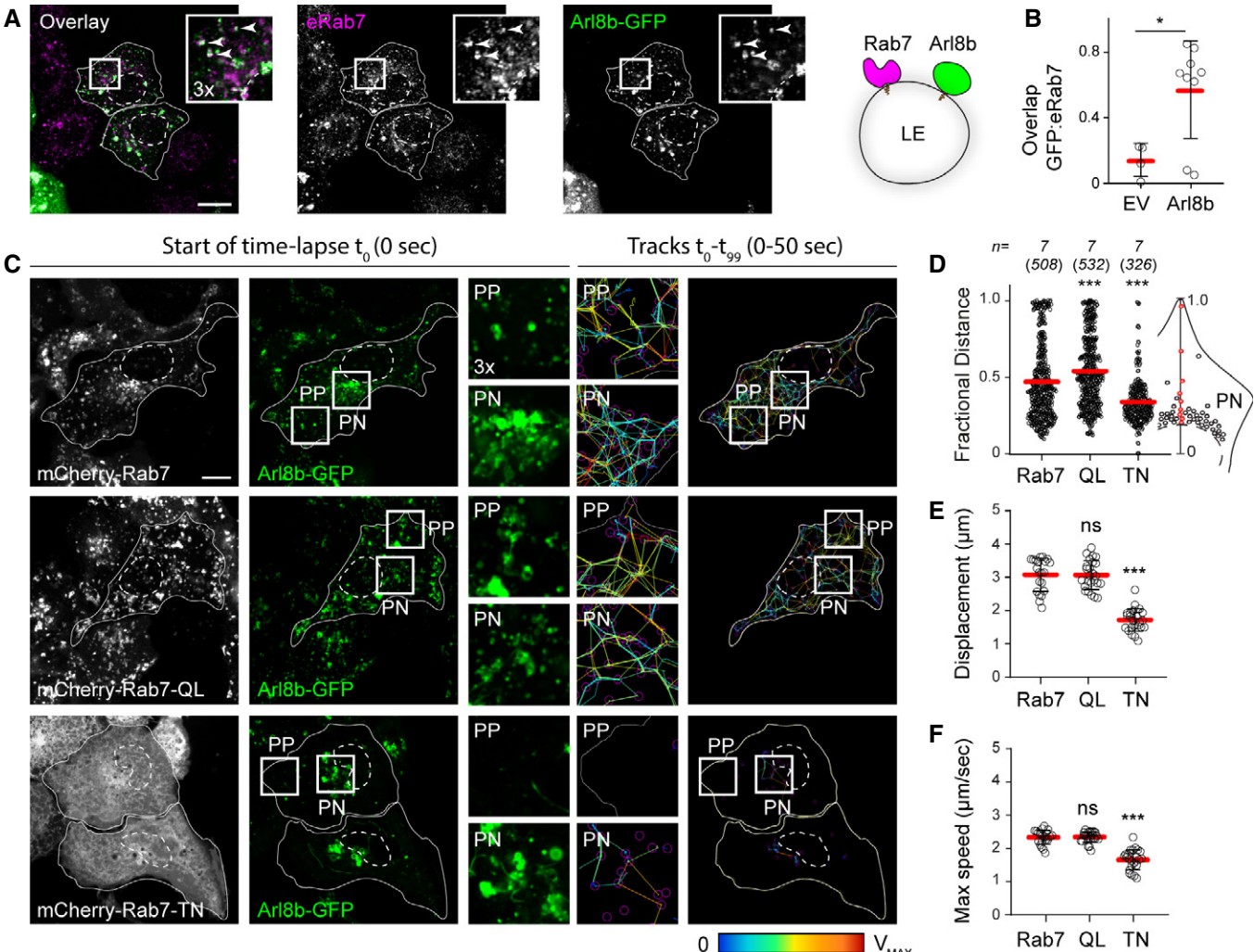

**Figure 1.   Rab7 activity influences organization and motility of the Arl8b compartment.**

A, B   Hybrid Rab7/Arl8b compartment. (A) Representative confocal images of fixed HeLa cells expressing Arl8b-GFP (*green*), immunolabelled against endogenous Rab7 (eRab7, *magenta*). Zoom insets (3×) highlight select regions of colocalization (*white*), and white arrowheads point to vesicles positive for both GTPases. (B) Colocalization (Mander's overlap) of endogenous Rab7 with Arl8b-GFP versus free GFP (EV), $n_{EV}$ = 4, $n_{Arl8b}$ = 9 images (3 ≥ cells per image) analysed from 2 independent experiments. Significance: two-tailed Student's *t*-test, *$P$ < 0.05.

C–F   Analysis of Arl8b compartment organization and dynamics as a function of Rab7 activity status. (C) *Left and middle panels*: representative confocal images of live HeLa cells expressing mCherry-Rab7 or its mutants Q67L and T22N (*white*), together with Arl8b-GFP (*green*), taken at the start of time-lapse ($t_0$). *Right panels*: tracks followed by Arl8b-positive vesicles during the time-lapse lasting 50 s recorded at 0.5 s per frame, with highest displacement rates for each track depicted on a rainbow colour scale (blue: immobile; red: maximum mobility per time interval). Zoom insets (2.8×) highlight select peripheral (PP) and perinuclear (PN) cell regions (see also Movies EV1–EV3). (D) Plot of Arl8b-positive pixel distribution expressed as fractional distance along a straight line from centre of nucleus (0) to the plasma membrane (1.0), numbers of (pixels) plotted given above each scatter, $n$ = 7 cells analysed per condition from 2 independent experiments. (E, F) Quantification of mean Arl8b vesicle displacement and maximum speed, respectively, $n_{Rab7}$ = 21, $n_{QL}$ = 25, $n_{TN}$ = 24 images (3 ≥ cells per image) analysed from 2 independent experiments. Significance: one-way ANOVA test (relative to wild type Rab7), ***$P$ < 0.001, ns: not significant.

Data information: Cell and nuclear boundaries are demarcated with solid and dashed lines, respectively, all scale bars: 10 μm. Graphs report mean (red line) of sample values (open circles), and error bars reflect ± SD.

modulated specific transport routes through relevant effector molecules. Overexpression of the Rab7 effector RILP, responsible for recruitment of the dynein motor complex (Jordens *et al*, 2001), yielded acute perinuclear (PN) clustering of Rab7-positive endosomes (Fig 2A–C), rendering the cell periphery (PP) devoid of these structures. In the same cells, Arl8b-positive vesicles also exhibited PN accumulation in the presence of RILP (Fig 2A–C). On the other hand, introduction of the Arl8b effector SKIP for recruitment of plus

end-directed motor kinesin-1 (Rosa-Ferreira & Munro, 2011) resulted in a concerted redistribution of Arl8b-positive endosomes to the periphery, with Rab7 also following suit (Fig 2A–C). This ability of Rab7 and Arl8b to respond to the other's effector engagement suggested that the hybrid compartment is sensitive to demands exerted on both GTPases.

When both effectors were co-expressed in the same cells, segregation of Arl8b/SKIP from Rab7/RILP was observed

(Fig 2A–D), respectively, splitting the late compartment between PP and PN regions of the cell (Fig EV1A–C). Such segregation was not a result of mere activation of opposing transport machineries, as demonstrated by co-expression of RILP with FYCO1—an effector of Rab7 for plus-end-directed LE transport

(Pankiv *et al*, 2010; Figs 2A–C and EV1A–C). Instead, when Rab7-positive endosomes were forced to move in both directions simultaneously, intermediate distribution of the late compartment was observed, illustrating a tug-of-war principle (Soppina *et al*, 2009; Hancock, 2014). Furthermore, engagement of Arl8b with

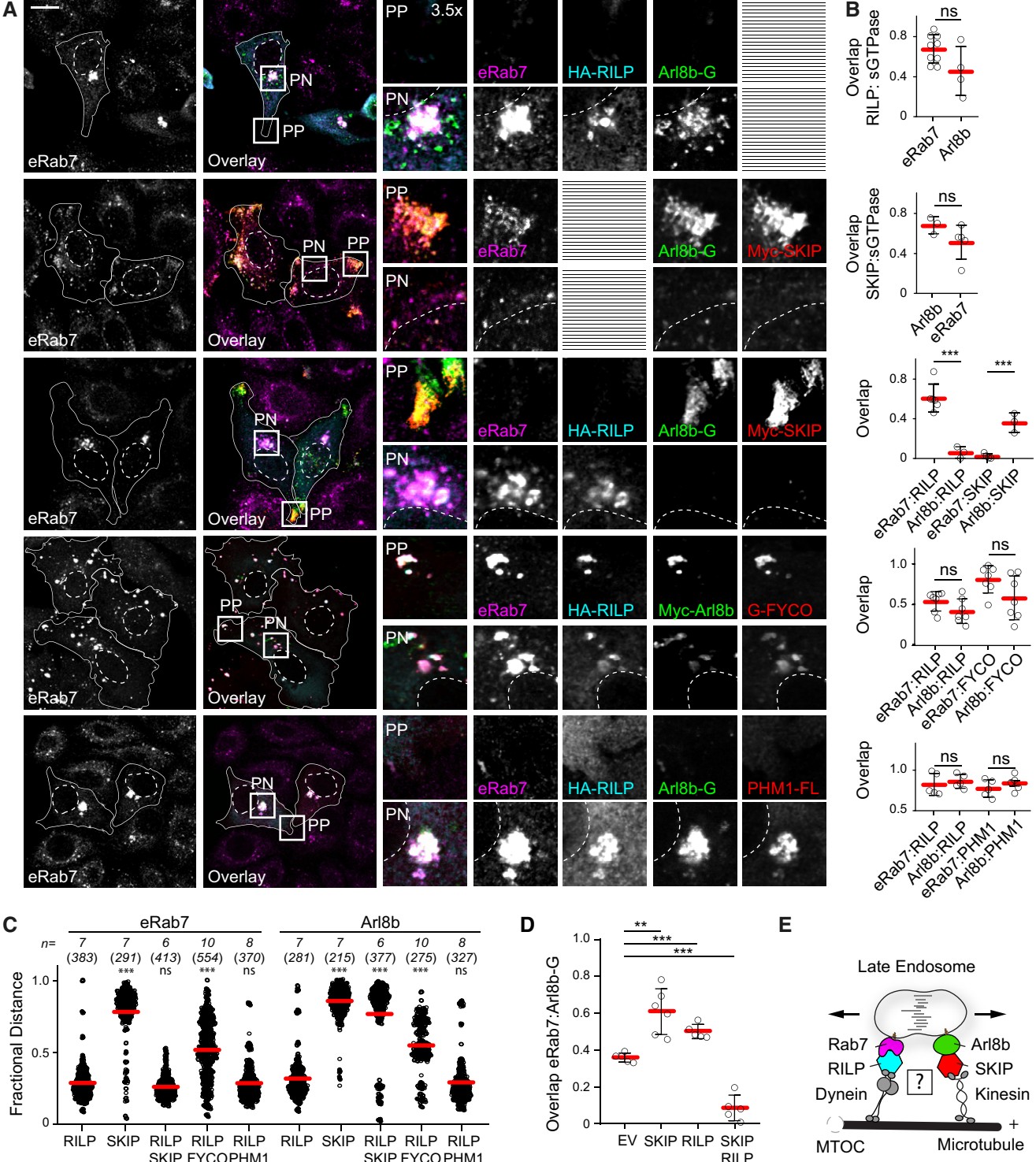

**Figure 2.**

◄

**Figure 2. Rab7 and Arl8b mediate spatial segregation of late compartments via their effectors RILP and SKIP.**

A–D   Effects of select transport route activation on the spatial organization of late compartments. (A) Representative confocal images of fixed HeLa cells ectopically expressing Arl8b-GFP or Arl8b-Myc (*green*), in combination with HA-RILP (*cyan*) and/or Myc-SKIP, PLEKHM1-FLAG or GFP-FYCO1 (*red*), immunolabelled against endogenous Rab7 (eRab7, *magenta*) and the indicated epitope tags. Cell and nuclear boundaries are demarcated with solid and dashed lines, respectively, and zoom insets (3.5×) highlight select peripheral (PP) and perinuclear (PN) cell regions, scale bar: 10 μm. (B) Colocalization (Mander's overlap) between the indicated pairs of proteins, $n \geq 3$ images ($3 \geq$ cells per image) analysed per condition from 2 independent experiments. (C) Plot of eRab7- or Arl8b-GFP/Myc-Arl8b-positive pixel distribution in response to the indicated effector perturbations, expressed as fractional distance along a straight line from centre of nucleus (0) to the plasma membrane (1.0), number of (pixels) plotted given above each scatter, $n \geq 6$ cells per condition analysed from 2 independent experiments. Significance: one-way ANOVA test (relative to RILP only), ***$P < 0.001$, ns: not significant. (D) Colocalization (Mander's overlap) between Arl8b and Rab7 as a function of the indicated effector perturbations, $n \geq 5$ images ($3 \geq$ cells per image) analysed per condition from 2 independent experiments.
E    Graphical summary of late compartment segregation as mediated by Rab7 and Arl8b.

Data information: Graphs report mean (red line) of sample values (open circles), and error bars reflect ± SD. Unless stated otherwise, significance was assessed using 2-tailed Student's *t*-test, **$P < 0.01$, ***$P < 0.001$, ns: not significant (See also Fig EV1A–C).

PLEKHM1 instead of SKIP could not produce vesicle segregation in the presence of RILP (Figs 2A–C and EV1A–C), presumably due to lack of kinesin-interacting determinants in the resulting complex. Hence, spatial compartmentalization of the late endolysosomal repertoire is predicated on co-activation of opposing transport routes directed by different GTPases (Fig 2E), implicating RILP and SKIP in the maintenance of endolysosomal membrane identities.

To evaluate whether the PN/PP segregation observed upon co-activation of Rab7-driven transport to the microtubule minus-end and Arl8b-mediated plus-end-directed movement yields distinct sub-populations of organelles, we examined ultrastructural characteristics of affected compartments. To aid in detection of relevant organelles, HeLa cells, wherein CD63 was endogenously tagged with GFP, were used in combination with SiR-Lysosome staining (Fig 3). In unperturbed cells, analysis of the peripheral cytoplasm revealed predominantly MVBs harbouring homogeneous ILV contents, while later organelles, such as multilamellar lysosomes and endolysosomes exhibiting both multivesicular and multilamellar characteristics were more prevalent in the perinuclear region (Fig 3A). These observations are consistent with an earlier report demonstrating peripherally located late compartments to be less acidic and proteolytic relative to their perinuclear counterparts (Johnson *et al*, 2016).

Ultrastructural examination of the over-crowded PN region in cells ectopically expressing RILP revealed heterogeneous endolysosomal profiles, many of which appeared enlarged and harboured abnormal intraluminal contents (Fig 3B). Furthermore, neither canonical MVBs nor multilamellar lysosomes were appreciably observed in this condition (Fig 3B), suggesting that hyper-fusion between these morphologically distinct organelles may have taken place, giving rise to aberrant hybrid states. By contrast, in the presence of ectopically expressed SKIP, normal MVBs were preserved (Fig 3C). These organelles localized predominantly to the leading edge of vesicles migrating to cell tips in response to activation of plus-end-directed movement by SKIP (Fig 3C), implying that they constitute preferred targets for Arl8b/SKIP-mediated transport over their later counterparts (i.e. endolysosomes). Importantly, addition of SKIP in the background of RILP overexpression rescued normal MVB profiles, which, similar to the control situation, were located primarily in the cell periphery (Figs 3D and EV1D, Movies EV4 and EV5). These results indicate that a balance between the activities of Rab7 and Arl8b helps maintain a healthy MVB repertoire.

**SKIP is a negative effector of Rab7**

Based on the evidence discussed above, we hypothesized that SKIP targets a subset of Rab7-positive membranes for transport to the cell periphery. Indeed, ultrastructural analysis confirmed that both proteins can occupy the same MVB membrane (Fig 4A) and co-isolate from cells (Fig 4B). Furthermore, the interaction between SKIP and Rab7 was strengthened by the constitutively active Rab7-Q67L (Fig 4B and C). This enhancement in binding was not observed for either RILP or PLEKHM1 under the same reaction conditions (Figs 4B and EV1E), suggesting that the interaction with SKIP is likely more transient. Importantly, binding of SKIP (like that of RILP and PLEKHM1) was inhibited by Rab7-T22N (Figs 4B and C, and EV1E), indicating that SKIP prefers the active form of Rab7—a behaviour characteristic of an effector. We therefore tested whether the interaction between SKIP and Rab7 is direct. *In vitro* precipitation assays demonstrated that SKIP is able to bind recombinant GST-Rab7, albeit to a lesser extent than GST-Arl8b (Figs 4D and EV1F). Truncation analysis (Fig 4E) demonstrated that binding of SKIP to Rab7 does not involve the N-terminal RUN domain responsible for contacting Arl8b (Rosa-Ferreira & Munro, 2011). Instead, the C-terminal half of SKIP, spanning amino acids 537–1,019, mediates the interaction with Rab7, and removal of either residues 874–1,019 (construct 537–873) or 537–744 (construct 745–1,019) is detrimental to the interaction (Figs 4E and EV1H). Furthermore, alignment of the SKIP sequence against that of established Rab7 effectors revealed the presence of a canonical Rab7-interacting KML motif (McEwan *et al*, 2015b) at positions 610–612, with a conserved substitution of L for I (Fig 4E), and mutation of this KMI motif to AAA reduced the interaction between SKIP and Rab7 by roughly half (Fig 4F and G). Taken together, these findings demonstrate that SKIP binds Rab7 in a manner consistent with that of a bona fide effector and suggest that a single SKIP molecule could interact with both Rab7 and Arl8b (Fig 4H), thus setting the stage for a negotiation between them.

To investigate whether SKIP associates with Rab7-positive membranes prior to instigating plus-end-directed transport we designed a system wherein a pre-existing cytosolic pool of SKIP could be induced onto membranes. GFP-SKIP expression construct was fused to an oestrogen receptor (ER) fragment, allowing retention of the ER-GFP-SKIP chimera in the cytosol through an interaction with HSP90 (Vigo *et al*, 1999; Rabinovich *et al*, 2008). Addition of tamoxifen would then release SKIP from HSP90, recovering its intended localization and function, thereby bringing SKIP-mediated

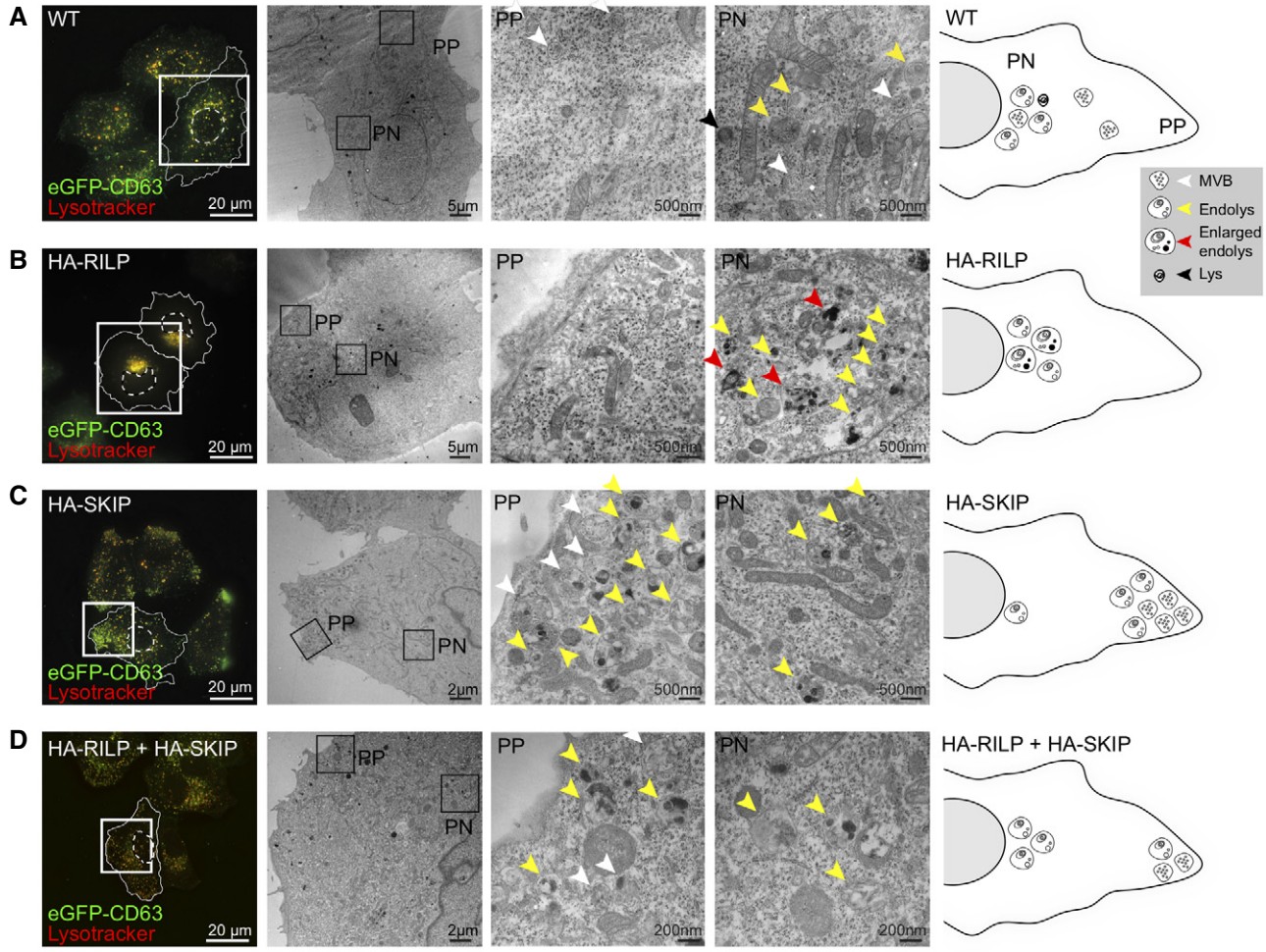

**Figure 3. Ultrastructural characterization of late compartment segregation along the perinuclear/peripheral axis.**

A–D Correlative light and electron microscopy (CLEM) on (A) untransfected (WT) HeLa cells harbouring endogenous CD63 labelled with GFP as compared to those ectopically expressing (B) HA-RILP, (C) HA-SKIP or (D) both HA-RILP and HA-SKIP. *Left panels*: wide-field fluorescence images of fixed cells showing endogenous GFP-CD63 (*green*) and SiR-lysosome-stained endosomes and/or lysosomes (*red*). Cell and nuclear boundaries are demarcated with solid and dashed lines, respectively, and zoom insets highlight regions selected for EM imaging. *Middle panels*: overview electron micrographs of perinuclear (PN) and peripheral (PP) cell regions selected for further analysis. Various endolysosomal subtypes are designated by arrowheads: MVBs (*white*), lysosomes (*black*), endolysosomes (*yellow*) and abnormal/enlarged endolysosomes (*red*). *Right panels*: graphical representations of endosomal distribution under the indicated conditions based on ultrastructural characterization. Scale bars as indicated (see also Fig EV1D, and Movies EV4 and EV5).

transport under chemical control for timed release (Fig 4I). In the absence of tamoxifen, no stable targeting of ER-GFP-SKIP to endosomal membranes was observed (Fig 4J, Movie EV6). By contrast, within minutes following tamoxifen addition, ER-GFP-SKIP could be readily detected on mCherry-Rab7-positive vesicles, while no appreciable SKIP compartment had yet been generated at cell tips (Fig 4K). Over time, SKIP-positive vesicles increasingly accumulated at cell tips (the end points of plus-end-directed transport), while the population of Rab7-positive endosomes existing outside of the perinuclear cluster diminished (Fig 4K, Movies EV7 and EV8). These observations demonstrate that SKIP actively selects Rab7-positive membranes before initiating transport to the cell periphery.

Having shown that SKIP and RILP both target Rab7 and collaborate in segregation of late compartments between Rab7 and Arl8b,

we tested whether loss of these effectors would result in the expansion of a hybrid compartment. Indeed, depletion of either RILP or SKIP augmented colocalization between endogenous Arl8b (tagged with GFP) and Rab7 (Fig 5A–C), complementing the overexpression studies presented in Fig 2. Given that SKIP interacts with both GTPases, but in the presence of RILP transports only Arl8b-positive endosomes to the cell periphery, we considered whether SKIP induces removal of Rab7 to ultimately achieve qualitative segregation of late compartments. To test this, we examined the effect of Rab7 activity on the identity of SKIP-positive vesicles. While ectopic expression of wild type GFP-Rab7 afforded efficient segregation between RILP and SKIP sub-compartments, expression of either constitutively active or inactive Rab7 mutants inhibited this process (Fig 5D and E). In the former perturbation, Rab7-Q67L and RILP

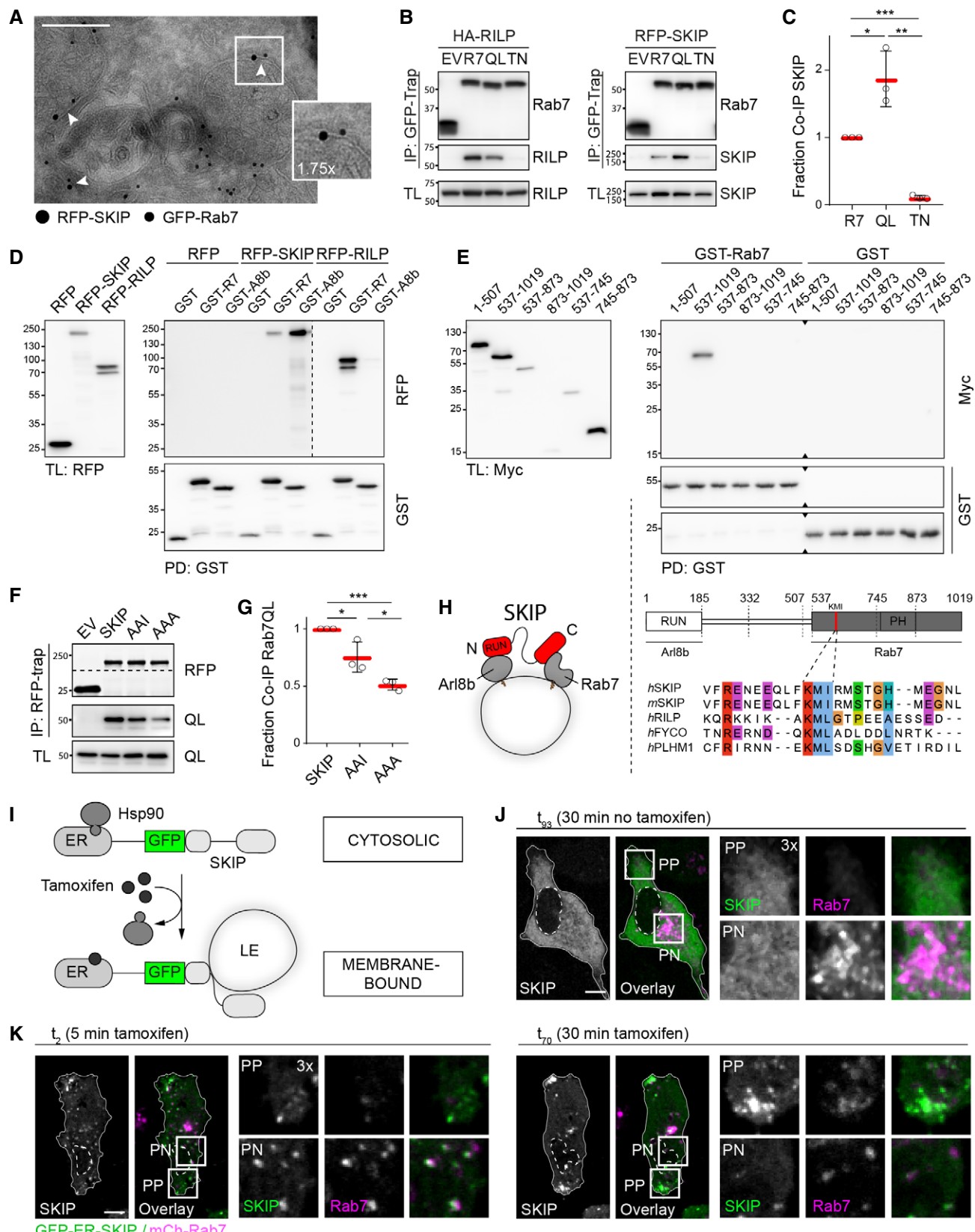

**Figure 4.**

**Figure 4. SKIP interacts with Rab7 through a canonical effector motif.**

A   Electron micrograph of sections immunolabelled against RFP-SKIP (15 nm gold) and GFP-Rab7 (10 nm gold). Arrowheads and zoom inset (1.75×) highlight presence of RFP-SKIP and GFP-Rab7 on the same endosomal membrane, scale bar: 200 nm.
B   Co-immunoprecipitations (Co-IP) of HA-RILP and RFP-SKIP with GFP-Rab7 (R7) versus its mutants Q67L (QL) and T22N (TN) from HEK293T cells using GFP-trap beads. Representative immunoblots against GFP, HA and RFP are shown, EV: empty vector, IP: immunoprecipitation, TL: total lysate (see also Fig EV1E).
C   Quantification of interaction between SKIP and Rab7 mutants expressed as fraction Co-IP relative to wild-type Rab7, n = 3 independent experiments.
D, E   In vitro glutathione precipitation assays. (D) Pull-down (PD) of RFP-SKIP or RFP-RILP from HEK293T cell lysates using recombinant GST-Rab7 versus GST-Arl8b and free GST. Representative immunoblots against RFP and GST are shown (see also Fig EV1F). (E) SKIP truncation analysis by PD against GST-Rab7. Top panels: representative immunoblots against Myc and GST (see also Fig EV1G). Bottom panels: schematic representation of SKIP domain organization. Regions of SKIP capable of interacting with Arl8b versus Rab7 are demarcated with solid black lines. An alignment of human (h) and murine (m) SKIP sequences to known effectors of Rab7 surrounding the conserved KML/I effector motif at residues 610–612 of SKIP is provided.
F, G   Co-IP of RFP-SKIP versus its KMI motif mutants AAI and AAA with constitutively active GFP-Rab7 Q67L using RFP-trap beads (see also Fig EV1H). (F) Representative immunoblots against GFP and RFP. (G) Quantification of interaction between SKIP mutants with Rab7 expressed as fraction Co-IP relative to wild-type SKIP, n = 3 independent experiments.
H   Graphical summary of SKIP as a dual effector of Arl8b and Rab7.
I–K   Time-lapse of SKIP-mediated transport of late endosomes. (I) Schematic representation of tamoxifen-induced activation of SKIP onto endosomal membranes. (J, K) Live HeLa cells co-expressing GFP-ER-SKIP (green) and mCherry-Rab7 (magenta) together with HA-RILP (unstained) expressed at low levels (cells transfected at 1:5 RILP:SKIP ratio) were imaged in the (J) absence or (K) presence of tamoxifen, allowing on-demand association of SKIP with endosomal membranes. Confocal frames from time-lapses taken at the indicated time points following treatment are shown. Cell and nuclear boundaries are demarcated with solid and dashed lines, respectively, and zoom insets (3×) highlight select peripheral (PP) and perinuclear (PN) cell regions, scale bars: 10 μm (see also Movies EV6–EV8).

Data information: Graphs report the mean (red line) of sample values (open circles), error bars reflect ± SD. All significance was assessed using 2-tailed Student's t-test: *P < 0.05, **P < 0.01, ***P < 0.001.
Source data are available online for this figure.

were retained on peripheral SKIP-positive vesicles, while in the latter case, due to the dominant negative quality of Rab7-T22N, the Rab7/RILP compartment was absent, and no segregation between RILP and SKIP could be observed (Fig 5D and E). These data implicate GTPase activity status of Rab7 in the compartmentalization of the endolysosomal repertoire and suggest that inactivation and removal of Rab7 from the endosomal membrane takes place with commencement of SKIP-associated transport (Fig 5F).

## GAP TBC1D15 removes Rab7 from membranes designated by SKIP

Inactivation of Rab GTPases is typically facilitated by cognate GAP molecules in possession of Tre-2/Bub2/Cdc16 (TBC) domains (Fukuda, 2011). We hypothesized that loss of relevant GAP activity would maintain active Rab7 on structures selected by Arl8b/SKIP, thus giving rise to a tug-of-war phenotype characterized by breakdown of bilateral architecture and concomitant disorganization of transport. Three TBC family proteins previously described to possess GAP activity towards Rab7 were examined (Fig 6A): TBC1D2, implicated in ILV formation and (Carroll et al, 2013; Jaber et al, 2016); TBC1D5, associated with the retromer complex (Jimenez-Orgaz et al, 2018; Seaman et al, 2018); and TBC1D15, described to operate at endosome-mitochondria contact sites (Zhang et al, 2005). Depletion of TBC1D2 did not markedly affect intracellular distribution of the late compartments carrying CD63, while knockdown of either TBC1D5 or TBC1D15 perturbed the PP/PN balance (Fig 6B and C). The former scenario led to accumulation of CD63-positive structures at cell tips, reflecting an exaggerated PN/PP divide. By contrast, loss of TBC1D15 produced dispersion of late compartments (Figs 6B and C, and EV2A and B) throughout the cell (Fig 6D and E), abrogating the PN/PP dichotomy expected from a tug-of-war between Rab7 and Arl8b. This phenotype of TBC1D15 depletion could be partially rescued by re-expression of siRNA-resistant GFP-TBC1D15, but not GFP-TBC1D5, implying that the functions of these two GAPs are not interchangeable. Given these

considerations, we concluded that of the 3 known Rab7 GAPs, TBC1D15 is most likely to play a direct role at the Rab7/Arl8b interface, while TBC1D5 regulates a parallel pathway emanating from the same organelles.

To integrate TBC1D15 function within the cell biological context of Arl8b/Rab7 interplay, we tested the consequences of TBC1D15 insufficiency on organelle architecture and motility of late compartments. Ultrastructural examination revealed loss of normal MVB and lysosome morphologies in cells depleted of TBC1D15. Instead, late organelles were now comprised of enlarged endolysosomes harbouring aberrant luminal contents (Fig 6F), closely resembling alterations incurred upon overexpression of RILP (Fig 3B). Additionally, loss of TBC1D15 gave rise to disorganized movement of endolysosomes in cellular space (Fig 6G, and Movies EV9 and EV10), akin to that observed in cells expressing constitutively active Rab7 Q67L (Fig 1C). This was accompanied by a marked reduction in the fast-moving vesicle repertoire (Fig 6H), pointing to a pivotal role for TBC1D15 in the regulation of peripheral LE/Ly motility. In agreement with a previous report (Peralta et al, 2010), depletion of TBC1D15 reduced cycling of Rab7 on endosomal membranes (Fig 6I), indicating altered activity dynamics.

We next tested whether TBC1D15 influences partitioning of the endolysosomal pool between the PN cloud and the cell periphery upon co-activation of opposing transport routes downstream of Rab7 and Arl8b. Silencing TBC1D15, but not its homologue TBC1D17, led to redistribution of a substantial proportion of endogenous Rab7 pool to the tips of cells (PP) along with SKIP, while in control cells nearly all membrane-localized Rab7 remained in PN clusters marked by RILP (Figs 7A–D and EV2A–C). Notably, co-transport of RILP with endogenous Rab7, retained on peripheral structures marked by SKIP, was not observed under conditions of TBC1D15 loss (Fig 7A), remaining on perinuclear Rab7 instead. This observation echoes a protective phenomenon previously reported for other effector proteins when bound to their client GTPase (Nagelkerken et al, 2000) and imply that SKIP-associated TBC1D15

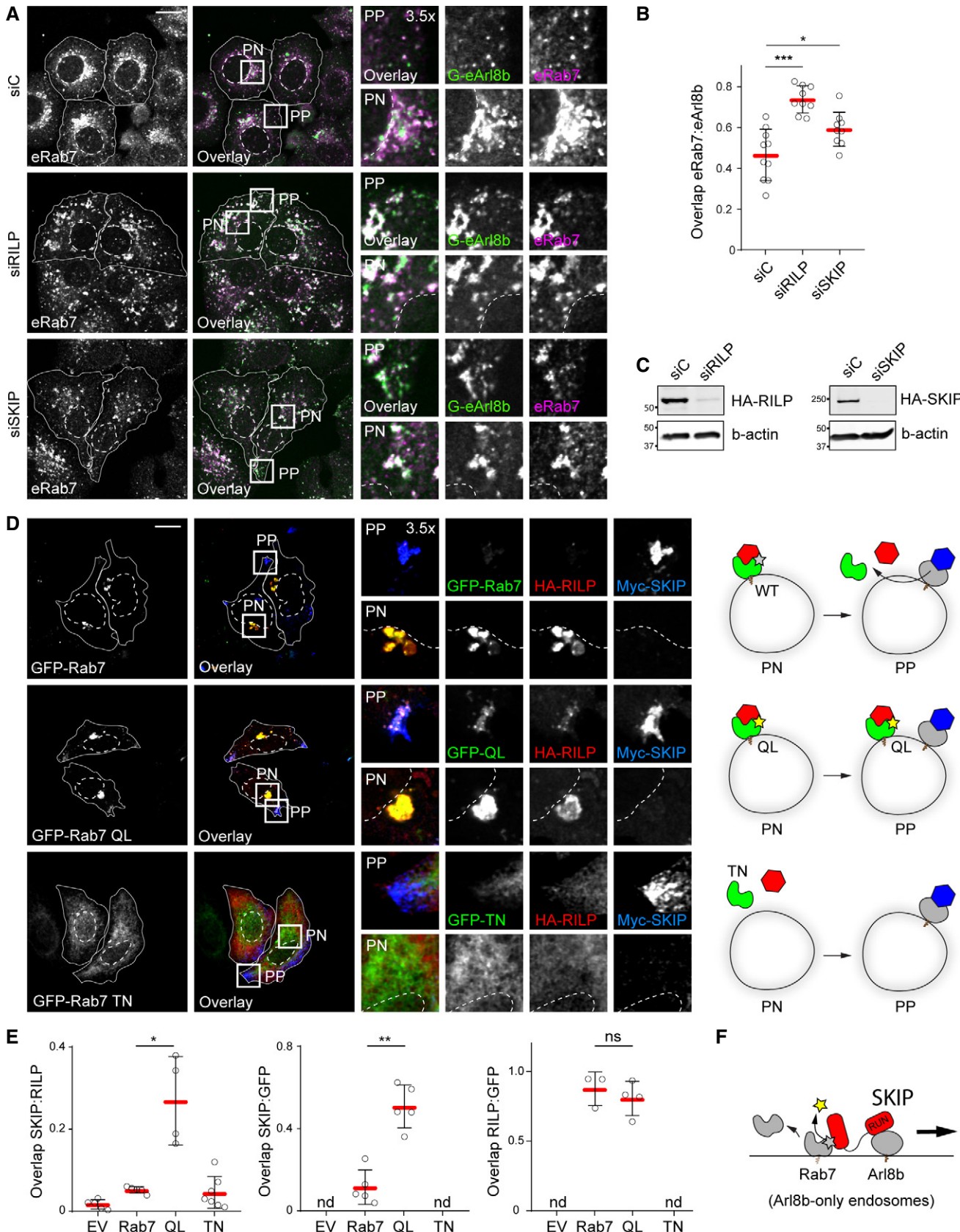

Figure 5.

**Figure 5. Late compartment segregation is predicated on removal of Rab7 from SKIP-positive membranes.**

A–C  Consequences of effector depletion on the endogenous Arl8b/Rab7 hybrid compartment. (A) Representative confocal images of fixed HeLa cells harbouring GFP-tagged endogenous Arl8b (G-eArl8b, green), transfected with the indicated siRNAs and immunolabelled against endogenous Rab7 (eRab7, magenta). (B) Colocalization (Mander's overlap) between endogenous Arl8b and Rab7 in response to effector depletion, $n_{siC} = 10$, $n_{siRILP} = 9$, $n_{siSKIP} = 9$ images (4 ≥ cells per image) analysed from 2 independent experiments. (C) Immunoblot analysis for depletion efficiency of SKIP and RILP, with actin as loading control.

D, E  Effect of Rab7 GTPase activity status on its association with the peripheral SKIP compartment. (D) Left panels: Representative confocal images of fixed HeLa cells expressing GFP-Rab7 or its mutants Q67L or T22N (green) together with HA-RILP (red) and Myc-SKIP (blue), immunolabelled against the indicated epitope tags. Right panels: Schematic overview per condition. (E) Colocalization (Mander's overlap) between the indicated protein pairs, $n_{Rab7} = 5$, $n_{QL} = 5$, $n_{TN} = 7$ images (2 ≥ cells per image) analysed from 2 independent experiments.

F  Graphical summary of Rab7 removal from the SKIP compartment.

Data information: Cell and nuclear boundaries are demarcated with solid and dashed lines, respectively, and zoom insets (3.5×) highlight select peripheral (PP) and perinuclear (PN) cell regions, scale bars: 10 μm. Graphs report the mean (red line) of sample values (open circles), error bars reflect ± SD. All significance was assessed using two-tailed Student's t-test: *$P < 0.05$, **$P < 0.01$, ***$P < 0.001$, ns: not significant, nd: not determined.

Source data are available online for this figure.

preferentially targets Rab7 not in complex with RILP. As before, depletion of TBC1D2 showed little effect in this context, while loss of TBC1D5 exhibited a similar phenotype to that of TBC1D15 (Figs 7B–D and EV2D), suggesting that the retromer/TBC1D5 pathway may compete with SKIP/TBC1D15 for the same Rab7 substrate. To ensure that the phenotypes observed above were not due to inactivation of another potentially relevant GTPase by TBC1D15, we conducted in vitro assays confirming that TBC1D15 specifically accelerates GTP hydrolysis of the GTPase domain belonging to Rab7, without affecting those of Rab5, Ran or Rab9 (Fig EV2E). Furthermore, Rab7 cargo MHC class II (MHC-II) was found to aberrantly traffic together with SKIP the absence of TBC1D15 (Figs 7E and EV2F), underscoring the notion that membrane dynamics at the Rab7 endolysosome are modulated at least in part though the interplay of SKIP and TBC1D15.

### SKIP recruits TBC1D15 not associated with mitochondria

Having shown that depletion of TBC1D15 hampers removal of Rab7 from the peripheral SKIP compartment, we tested whether SKIP is able to target this GAP to its membranes of choice. TBC1D15 localizes predominantly to the outer mitochondrial membrane (Onoue et al, 2013; Yamano et al, 2014) and has previously been shown to target Rab7 at membrane contact sites between mitochondria and endosomes (Wong et al, 2018). However, FRAP analysis performed on endogenous TBC1D15, N-terminally tagged with GFP in HeLa cells (Fig EV3A), revealed its association with mitochondrial membranes to be dynamic (Fig EV3B). Additionally, TBC1D15 negative for mitochondrial markers was found to colocalize with Rab7- and CD63-positive endosomes in unperturbed cells, as well as under depletion of the mitochondrial TBC1D15 anchor, FIS1 (Fig 8A and B), implying that TBC1D15 can be delivered to endosomes by other mechanisms. In support of this, FIS1 knockdown did not recapitulate key phenotypes of TBC1D15 loss of function, affecting neither the intracellular distribution of CD63-positive structures nor retention of Rab7 on the SKIP compartment (Figs 8C–F and EV3C).

Upon ectopic expression of SKIP, endogenous TBC1D15 could be detected at SKIP-positive vesicles located in the tips of cells, which did not co-label with either Tomm20 or MitoTracker (Figs 8G–J, and EV3D and E). On the other hand, TBC1D15 did not show appreciable colocalization with PLEKHM1 (Fig 8G and H), indicating

specificity in recruitment of this GAP to endosomal membranes. Taken together, these results demonstrate that TBC1D15 can be recruited to membranes selected by SKIP and imply that this acquisition does not involve direct contacts with mitochondrial (Fig 8K).

### TBC1D15 interacts with the HOPS complex to inactivate Rab7 from SKIP-positive membranes

To understand the molecular underpinnings of TBC1D15 recruitment to SKIP for inactivation of Rab7, we first sought to place SKIP/Rab7 interactions in the context of the greater Arl8b/SKIP transport complex. Both Arl8b and SKIP have been shown to recruit the HOPS complex to late endosomes by directly contacting VPS41 and VPS39 subunits, respectively (Khatter et al, 2015). Co-precipitation analysis of SKIP truncations mapped the interaction with VPS39 onto the central region of SKIP, overlapping the KMI motif, while its association with VPS41 extended further towards SKIP's N-terminus (Fig EV4A and B). This in turn suggested that engagement of Rab7 may influence acquisition of the HOPS complex by SKIP. Indeed, SKIP-AAA exhibited strongly reduced affinity for VPS39, as compared to its wild-type counterpart (Fig 9A and B), resulting in failure to recruit VPS39 to endosomes (Fig 9C and D). At the same time, co-isolation with VPS41 and its recruitment were improved by SKIP-AAA (Figs 9A–C and EV4C). Importantly, the SKIP mutant lacking the RUN domain failed to localize to endosomes (Figs 9C and D, and EV4C), indicating that association with Arl8b is a prerequisite for SKIP function on these organelles. Taken together, these results suggest that SKIP-associated HOPS assembly relies on contacts to both of its cognate GTPases.

Strikingly, co-expression of VPS39 with SKIP caused a marked improvement in acquisition of endogenous TBC1D15 (Fig 9E and F), and similar effects were observed for ectopic expression of other HOPS subunits (Figs 9E–G and EV4D). Furthermore, in all cases, co-recruitment of endogenous core HOPS subunits VPS18 or VPS11 was also observed (Fig 9H and I), implying that the complex (rather than isolated subunits) is involved in recruitment of TBC1D15 to SKIP-positive membranes. As expected, no contribution from FIS1 was observed in this context (Figs 9G and EV5A), excluding direct contribution of mitochondria-bound TBC1D15.

We next examined whether TBC1D15 is a component of the SKIP-associated complex. Because only a faint interaction was detected between SKIP and TBC1D15 by conventional co-IP (Fig EV5B), we turned to the BioID approach (Roux et al, 2012)

allowing interactors of a protein of interest to be covalently labelled with biotin in cells prior to lysis. SKIP was adapted with the BioID variant of a promiscuous biotin ligase domain, and in the presence of exogenously supplied biotin, TBC1D15 was biotinylated by BioID-SKIP at levels far exceeding those afforded by the free BioID moiety under the same conditions (Fig 10A and B). In the same

experiment, specific labelling of endogenous VPS18 by BioID-SKIP was also observed, indicating that both VPS18 and TBC1D15 are in complex with SKIP. Similarly, endogenous VPS18 (and VPS11) were detected at the SKIP compartment together with TBC1D15 (Fig EV5C). Taking these results together with enhanced recruitment of TBC1D15 resulting from overabundance of HOPS subunits

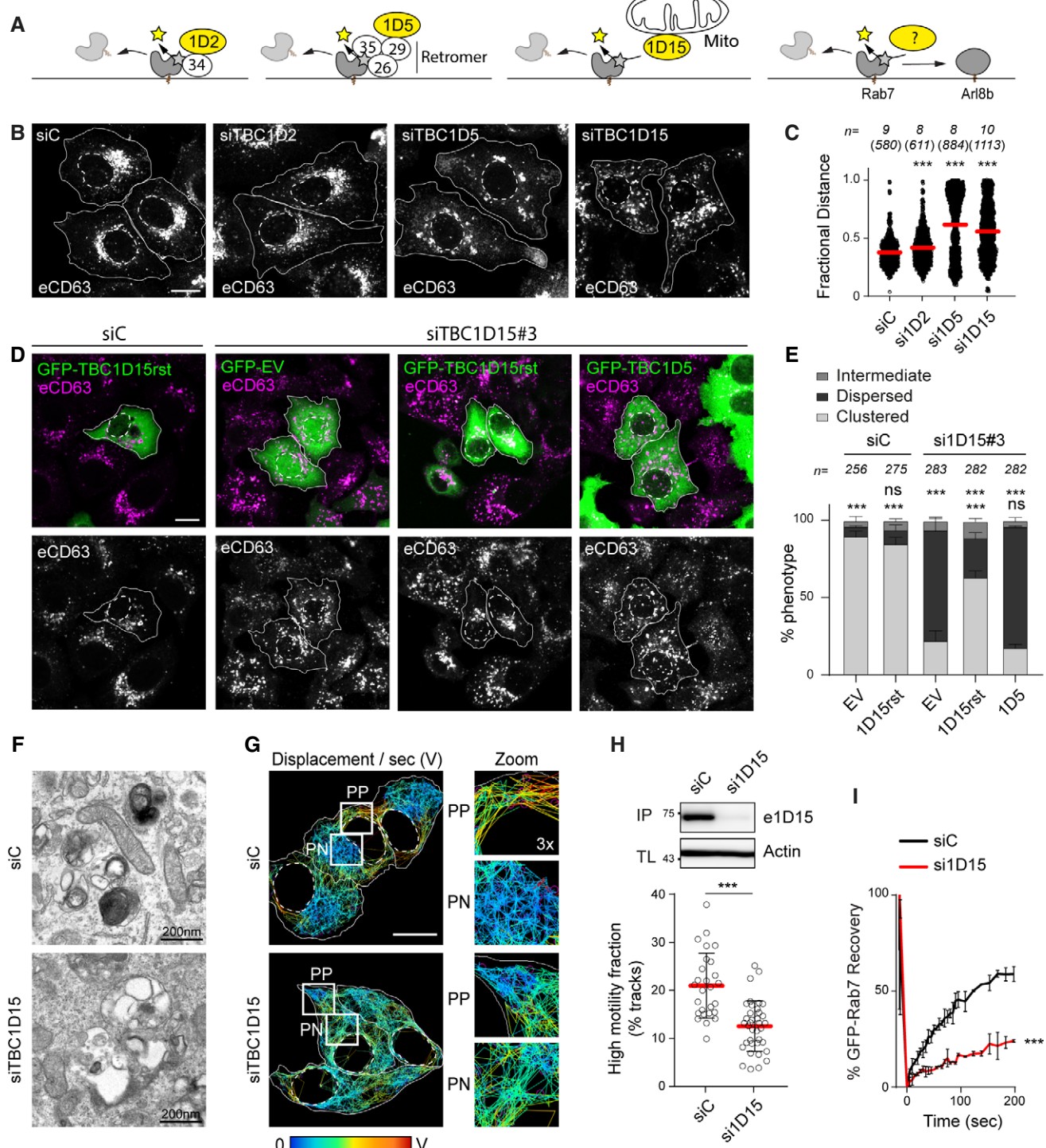

Figure 6.

**Figure 6.  Depletion of Rab7 GAP TBC1D15 disrupts the bilateral architecture, morphology and dynamics of late compartments.**

A–C  Identification of a Rab7 GAP that promotes endolysosomal system's bilateral architecture along the PN/PP axis. (A) Graphical summary depicting the functions of 3 known TBC domain-containing GAPs for Rab7. (B) Representative confocal images of fixed HeLa cells depleted of TBC1D2, 1D5 or 1D15 using siRNA oligo pools and immunolabelled against CD63 (*white*), scale bar: 10 μm. (C) Plot of CD63 pixel distribution as a function of TBC1D2/1D5/1D15 depletion expressed as fractional distance along a straight line from centre of nucleus (0) to the plasma membrane (1.0), number of (pixels) plotted given above each scatter, $n \geq 8$ cells per condition analysed from 2 independent experiments. Significance: one-way ANOVA test (relative to siC), ***$P < 0.001$ (see also Fig EV2A and B).

D, E  Rescue of TBC1D15 depletion phenotype. (D) Representative confocal images of fixed HeLa cells transfected with either control siRNA (siC) or oligo #3 targeting TBC1D15 (siTBC1D15) and ectopically expressing either GFP-EV, GFP-TBC1D5 or siRNA-resistant GFP-TBC1D15res (*green*), immunolabelled for CD63 (*magenta*), scale bar: 10 μm. (E) Quantification of rescue expressed as % cells (average) in the population exhibiting one of 3 phenotypes: clustered, dispersed or intermediate; total numbers of cells analysed per condition appear above each bar, $n = 3$ independent experiments. Significance (based on clustered phenotype): one-way ANOVA test relative to either siC/EV (*top row*) or si1D15/EV (*bottom row*), ***$P < 0.001$, ns: not significant.

F–H  Effects of TBC1D15 depletion on the morphology and dynamics of late organelles. (F) Representative electron micrographs of fixed HeLa cells transfected with either control siRNA (siC) or a pool of oligos targeting TBC1D15 (siTBC1D15) are shown, scale bars as indicated. (G) Analysis of late compartment dynamics as a function of TBC1D15 depletion. Tracks followed by SiR lysosome-positive vesicles during a time-lapse lasting 255 s (5 s per frame), with highest displacement rates for each track depicted on a rainbow colour scale (*blue*: immobile; *red*: maximum mobility per time interval). Zoom insets (3×) highlight select peripheral (PP) and perinuclear (PN) regions, scale bar: 10 μm (see also Movies EV9 and EV10). (H) Quantification of high motility fraction (% tracks with displacement rates above 0.9 μm/s), $n_{siC} = 27$, $n_{si1D15} = 37$ images ($2 \geq$ cells per image) analysed from 2 independent experiments. Effectiveness of TBC1D15 (si1D15) depletion is confirmed by immunoblot (also appearing in Fig EV2A) against endogenous TBC1D15 (e1D15) following its immunoprecipitation (IP), TL: total lysate.

I  FRAP of GFP-Rab7 in live HeLa cells transfected with either control siRNA (siC, *black line*) or a pool of oligos targeting TBC1D15 (si1D15, *red line*). Plotted is average GFP-Rab7 signal recovery during 200 s following bleaching, expressed as % of pre-bleach signal, $n = 3$ bleach regions per sample.

Data information: Cell and nuclear boundaries are demarcated with solid and dashed lines, respectively, as applicable. Graphs report mean (red line) of sample values (open circles), and error bars reflect ± SD. Unless stated otherwise, significance was assessed using two-tailed Student's *t*-test, ***$P < 0.001$.

Source data are available online for this figure.

(Fig 9F and G), we considered whether the HOPS complex interacts with TBC1D15, thereby bringing it in close proximity of SKIP. To this end, co-isolation of TBC1D15 with various HOPS members was assessed, demonstrating positive interactions with core subunits VPS16 and VPS18 (Fig 10C).

To evaluate whether the interaction with HOPS informs deposition of TBC1D15 onto vesicles selected by SKIP, we tested the effect of HOPS loss of function. Depletion of VPS18 markedly reduced distribution of TBC1D15 to the tips of cells along with SKIP (Fig 10D and E), a phenotype opposite of that observed with overabundance of VPS18 (Fig 9E and G). In agreement with the deleterious effect of VPS18 loss on TBC1D15 recruitment to SKIP, its depletion also reduced TBC1D15 biotinylation by BioID-SKIP (Fig 10F–H), indicating weakened interaction and hence substantiating a positive role for this HOPS subunit in recruitment of TBC1D15 to the SKIP complex.

Finally, to establish whether SKIP-associated TBC1D15 is functional against Rab7, we investigated Rab7 removal from target endosomes as a function of TBC1D15 GAP activity. In the presence of wild-type TBC1D15, virtually no Rab7 was detected on SKIP-positive endosomes located in cell tips (Figs 10I and EV5D). By contrast, TBC1D15 mutant R417A, deficient in stimulating GTP hydrolysis (Pan *et al*, 2006), failed to remove Rab7 from endosomal membranes selected by SKIP (Figs 10I and EV5D), despite being recruited to the SKIP/HOPS complex. Interestingly, PLEKHM1 could not mediate displacement of Rab7 by TBC1D15 under the same conditions (Figs 10I and EV5D), despite being able to attract the HOPS complex to the same degree as SKIP (Fig EV5C). Collectively, these observations support a model wherein TBC1D15 is specifically recruited to SKIP-positive membranes via the HOPS complex, and in doing so can access Rab7 present on the same membranes for its inactivation (Fig 10J). As a result, Rab7 is removed, allowing spatiotemporally resolved generation of the Arl8b/SKIP repertoire from the Rab7-positive compartment.

## Discussion

Maturation and motility of endosomes are coordinated by their ever-changing membrane-associated protein interactomes that mark different endosomal states. In particular, controlled arrivals and departures of GTPases to and from specific endosomal membranes determine the duration of time those endosomes persist in a given maturation state (*temporal* regulation) and inform their transport itineraries (*spatial* regulation). Most immediately, residence of specific GTPases on endosomal membranes is modulated by their cognate GEFs and GAPs, respectively, turning relevant GTPase activities "on" and "off". As such, control over the GTP hydrolysis cycle is instrumental in achieving regulated transitions from one GTPase to another, thereby timing endosomal maturation and controlling GTPase-directed transport. Until now, this type of GTPase handover mechanism on endosomes has been most extensively described for the Rab5-to-Rab7 conversion taking place during the critical early-to-late maturation step (Rana *et al*, 2015). In this study, we propose an additional handover mechanism occurring at the late endosomal stage—the Rab7-to-Arl8b switch. In this stepwise transfer, the GAP necessary to inactivate and remove Rab7 from mature MVBs is provided by the Arl8b/SKIP complex, thus inducing controlled disengagement of Rab7.

Because both Rab7 and Arl8b can mediate transport of vesicles towards the periphery, a question arises as to why the same endosomal compartment allows for two plus-end-directed transport machineries. Our observations suggest that installation of opposing transport machineries on *different* GTPases allows not only spatial but also qualitative (i.e. morphological) control over the LE/Ly repertoire. We characterize SKIP as a dual specificity effector, engaging Rab7 as well as Arl8b, which sets up a platform for a dialogue between them. This parallels what has recently been described for SKIP's family member PLEKHM1 (Marwaha *et al*, 2017). However, while PLEKHM1 bridged Arl8b and Rab7 residing on opposing membranes during fusion, resulting in a hybrid identity

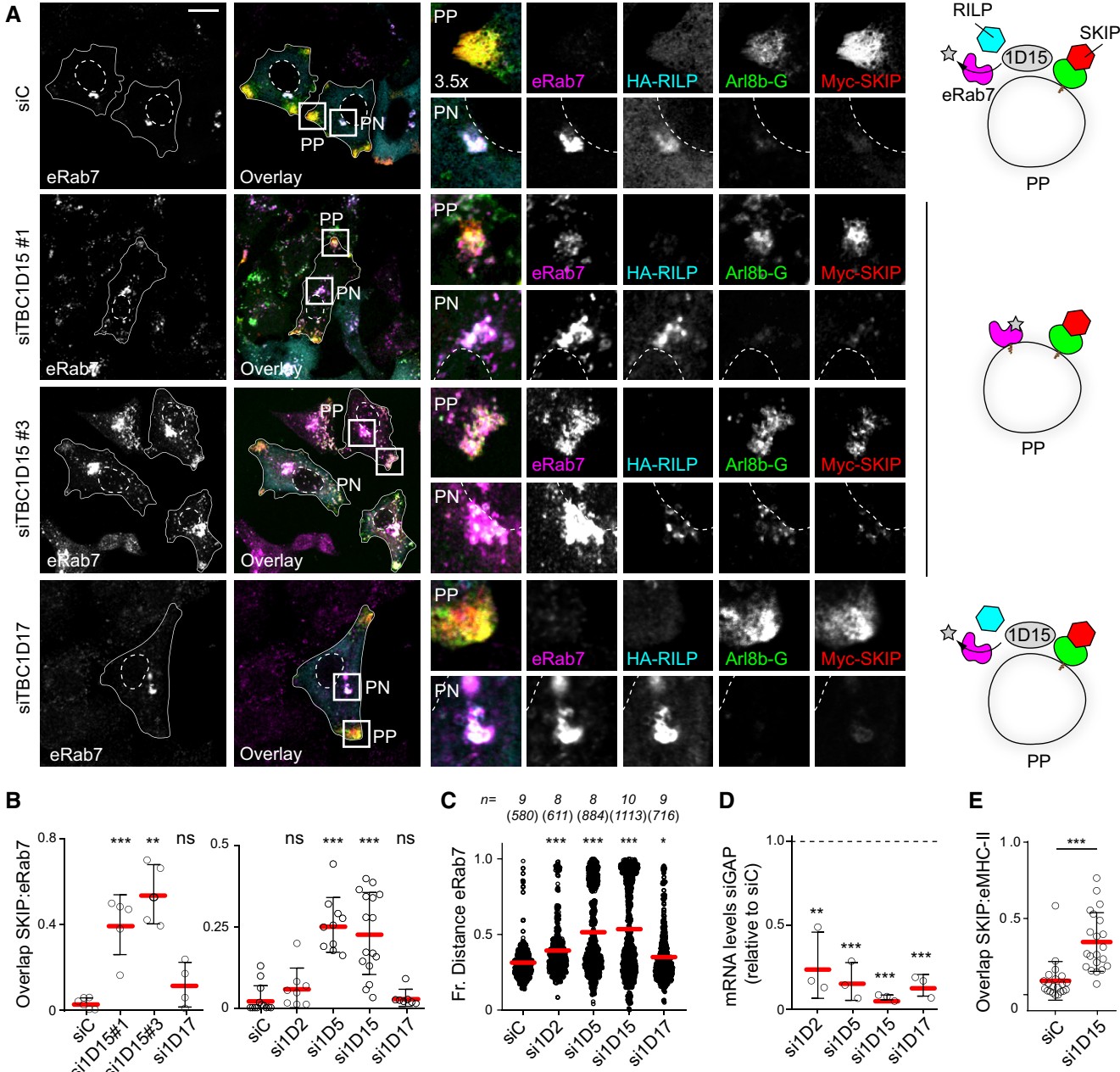

**Figure 7. GAP TBC1D15 facilitates spatial segregation of late compartments marked by Rab7 and Arl8b.**

A–D  Effects of depleting known GAPs for Rab7 on late compartment segregation mediated by RILP and SKIP. (A) Representative confocal images of fixed HeLa cells transfected with either control siRNA (siC), two different siRNA oligos targeting TBC1D15 (#1 and #3) or oligo pool targeting TBC1D17 and ectopically expressing GFP-Arl8b (*green*) in combination with HA-RILP (*cyan*) and Myc-SKIP (*red*), immunolabelled against endogenous Rab7 (eRab7, *magenta*) and the indicated epitope tags. Cell and nuclear boundaries are demarcated with solid and dashed lines, respectively, and zoom insets (3.5×) highlight select peripheral (PP) and perinuclear (PN) cell regions, scale bar: 10 μm. Graphical summaries appear on the right of each condition. (B) Colocalization of SKIP with endogenous Rab7 (Mander's overlap) in response to depletion of TBC-containing proteins, $n \geq 4$ images (3 ≥ cells per image) analysed per condition from 2 or more independent experiments (see also Fig EV2C). (C) Plot of Rab7-positive pixel distribution expressed as fractional distance along a straight line from centre of nucleus (0) to the plasma membrane (1.0), number of (pixels) plotted given above each scatter, $n \geq 8$ cells analysed per condition from 2 independent experiments. (D) Validation of depletion efficiencies expressed as fraction of the indicated mRNA remaining relative to siC assayed by qPCR, $n = 3$ independent experiments.

E  Effect of TBC1D15 depletion on Rab7-dependent MHC-II receptor trafficking to the SKIP-positive compartment, $n_{siC} = 21$, $n_{si1D15} = 21$ cells analysed from 3 independent experiments (see also Fig EV2F). Significance: two-tailed Student's *t*-test, ***$P < 0.001$.

Data information: Graphs report the mean (red line) of sample values (open circles), and error bars reflect ± SD. Unless stated otherwise, significance was assessed using one-way ANOVA test (relative to siC): *$P < 0.05$, **$P < 0.01$, ***$P < 0.001$, ns: not significant.

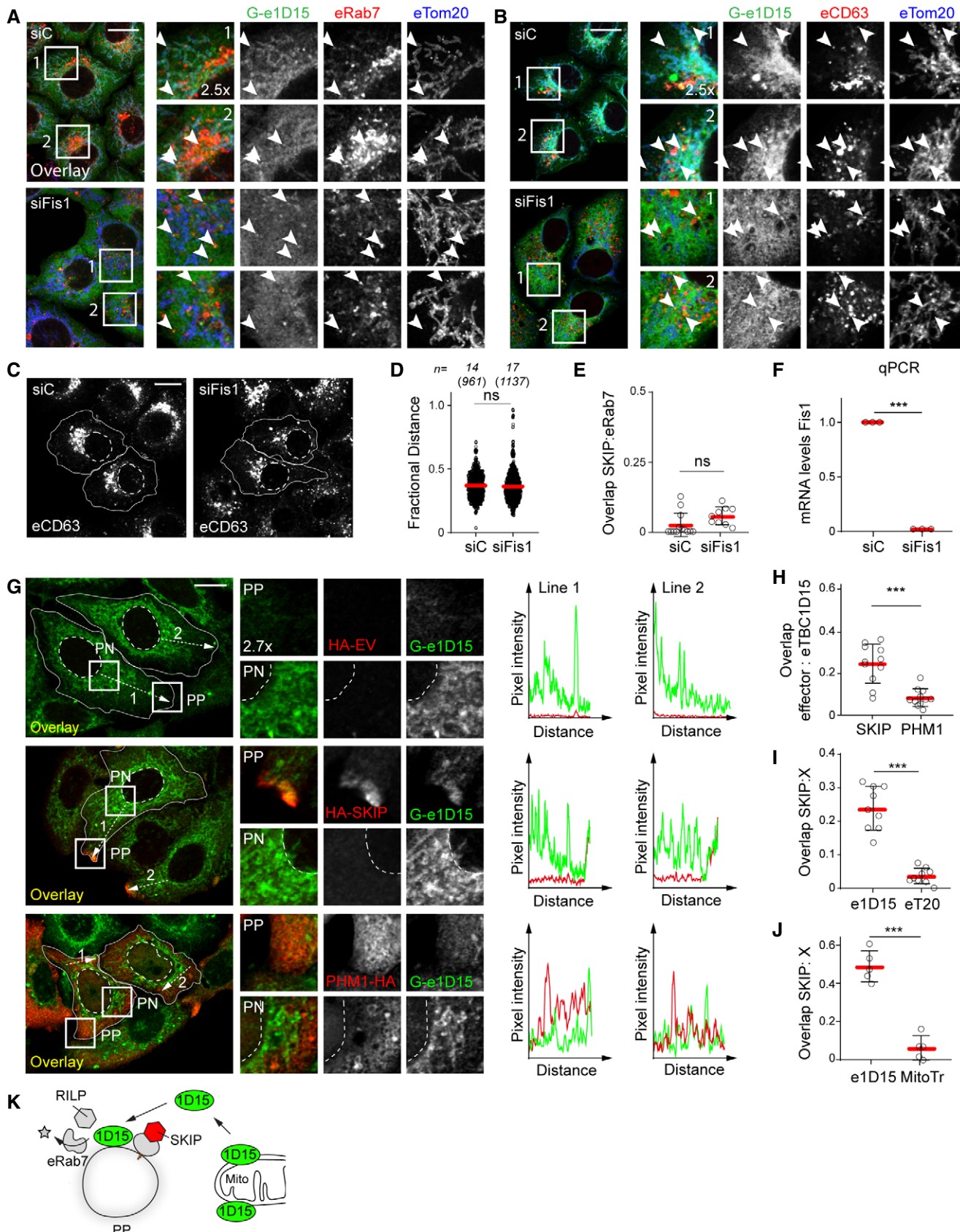

Figure 8.

◄

**Figure 8.  SKIP recruits non-mitochondrial TBC1D15.**

A, B   Representative confocal images of either control (siC) HeLa cells harbouring endogenous TBC1D15 tagged with GFP (Ge1D15, *green*) or those depleted of FIS1 using
       a pool of siRNA oligos, fixed and immunolabelled for the mitochondrial marker TOM20 (*blue*) in combination with either endogenous (A) Rab7 or (B) CD63 (*red*).
       Zoom insets (2.5×) highlight select cell regions, arrowheads point to vesicles positive for TBC1D15 and negative for TOM20 (*yellow*).
C–F   Effect of FIS1 on late compartment distribution. (C) Representative confocal images of fixed HeLa cells depleted of FIS1 using a siRNA oligo pool and
       immunolabelled against CD63 (*white*). (D) Plot of CD63 pixel distribution as a function of FIS1 depletion expressed as fractional distance along a straight line from
       centre of nucleus (0) to the plasma membrane (1.0), number of (pixels) plotted given above each scatter, $n_{siC}$ = 14, $n_{siFis1}$ = 17 cells analysed from 2 independent
       experiments. (E) Colocalization of SKIP with endogenous Rab7 (Mander's overlap) in response to FIS1 depletion, $n_{siC}$ = 13, $n_{siFis1}$ = 9 images (3 ≥ cells per image)
       analysed from 2 independent experiments (see also Fig EV2C). (F) Validation of FIS1 depletion efficiency assayed by qPCR and expressed as fraction FIS1 mRNA
       remaining relative to siC, n = 3 independent experiments.
G–J   Recruitment of endogenous TBC1D15 to the SKIP compartment. (G) *Left panels*: representative confocal images of fixed HeLa cells harbouring endogenous TBC1D15
       tagged with GFP (G-e1D15, *green*) and ectopically expressing either empty vector, HA-SKIP or PLEKHM1-HA (*red*), immunolabelled against HA. Zoom insets (2.7×)
       highlight select peripheral (PP) and perinuclear (PN) cell regions. *Right panels*: pixel plots of endogenous GFP-TBC1D15 (*green line*) and HA signals (*red line*)
       corresponding to the dashed white lines in (G). (H–J) Colocalization (Mander's overlap) between the indicated pairs of proteins, n ≥ 5 images, 3 ≥ cells per image,
       analysed per condition from 2 independent experiments (see also Fig EV3D and E).
K     Graphical summary of TBC1D15 recruitment to SKIP.

Data information: Cell and nuclear boundaries are demarcated with solid and dashed lines, where applicable, all scale bars: 10 μm. Graphs report mean (red line) of
sample values (open circles), and error bars reflect ± SD. All significance was assessed using two-tailed Student's *t*-test: \*\*\*$P$ < 0.001, ns: not significant.

compartment, SKIP likely engages these GTPases on the same membrane. The latter scenario ultimately leads to the removal of Rab7 from target membranes, restoring the Arl8b single identity repertoire. Delving deeper into the mechanism, we find that SKIP harbours a variant of the KML motif conserved in other Rab7 effectors and utilized to make direct contacts with the GTPase. In the case of SKIP, L is substituted by I, which may be responsible for the apparent lower affinity of Rab7 for this effector. This would be consistent with the proposed negative, and hence transient, effector function of SKIP towards Rab7 benefiting from easy disengagement. Curiously, the sequence context surrounding KMI residues in SKIP is closer to that of FYCO1, rather than for instance PLEKHM1 (McEwan *et al*, 2015b), suggesting that SKIP and FYCO1 may have evolved to compete for the same Rab7 substrate. What then determines which plus-end-directed pathway is chosen—and why— remains an open question. One obvious possibility is that the presence of Arl8b in close proximity to Rab7 skews the system towards SKIP, while its absence endows FYCO1 with the default plus-end effector status. Likewise, it is conceivable that formation of ER-endosome contacts organized by protrudin (Raiborg *et al*, 2015) would instead be promotive of the FYCO1-dependent transport route. How these and other mechanisms impinging upon transport route selectivity by late endosomes are integrated to support orderly spatiotemporal regulation of these complex organelles remains to be investigated.

For the Rab7-to-Arl8b switch to commence, Arl8b needs to either be activated and recruited to Rab7-positive structures, or already be found on the same membrane due to a prior fusion event (Marwaha *et al*, 2017). Subsequently, Arl8b-associated molecular machinery must come in contact with Rab7 to deliver the GAP for its inactivation and release. The first action is known to be mediated by the BORC complex, which is reported to function as a GEF for Arl8b (Pu *et al*, 2015). Interestingly, BORC shares a number of subunits with BLOC1, the complex reported to induce removal of Rab5 through the recruitment of the cognate GAP Msb3. Overlap in subunit build-up is not uncommon in remodelling of membrane identity, as illustrated by the VPS core shared between the CORVET and HOPS tethering complexes operating on early and late endosomes, respectively (Balderhaar & Ungermann, 2013). Because the HOPS complex can be recruited by both Rab7 and Arl8b (van der Kant

*et al*, 2015; Khatter *et al*, 2015), and activation of Arl8b by BORC has been shown to promote acquisition of HOPS by autolysosomes (Jia *et al*, 2017), HOPS constitutes a common factor ideally positioned to facilitate an ordered transition from Rab7 to Arl8b. Indeed, we find that the interaction between SKIP and Rab7, established through the KMI motif, supports acquisition of VPS39 by the SKIP-associated HOPS. Interestingly, the yeast homologue of VPS39 functions as a GEF for the Rab7 homologue Ypt7, but this attribute is not conserved in mammals (Peralta *et al*, 2010). Our observations suggest that in higher organisms this HOPS subunit has taken on the role of a sensor of Rab7 in macromolecule complexes assembled on endosomal membranes. Additionally, we find that in the absence of Rab7 contacts, SKIP exhibits higher affinity for other HOPS subunits, including VPS41 provided by Arl8b (Khatter *et al*, 2015), suggesting that SKIP and the associated HOPS complex straddle the two GTPases, poised for timely delivery of the Rab7 GAP.

In this study, we implicate the GAP TBC1D15 in the second step of the Rab7-to-Arl8b conversion—the inactivation and release of Rab7 from endosomal membranes selected by SKIP. TBC1D15 is not the only described GAP with biochemical activity against the Rab7 GTPase (Seaman *et al*, 2009; Frasa *et al*, 2010), and GAP molecules in general are known to operate with a certain degree of promiscuity (Fukuda, 2011). Therefore, GAP specificity is unlikely to rely exclusively on its catalytic activity, but rather depends on controlled targeting of said activity to the correct location at the appropriate time. For instance, TBC1D2A (Armus) has been shown to inactivate Rab7 during lysosomal maturation and starvation-induced autophagy (Carroll *et al*, 2013; Jaber *et al*, 2016), while TBC1D5 functions with the retromer complex to remove Rab7 participating in endosome-to-Golgi transport as well as in early steps of autophagy (Seaman *et al*, 2009; Popovic & Dikic, 2014). Our findings suggest that TBC1D5, but not TBC1D2, may compete with the SKIP/TBC1D15 pathway for the same Rab7 substrate, as loss of this GAP produces a phenotype akin to hyper-segregation of late compartments observed with overabundance of SKIP and RILP. In addition to other Rab7 GAPs, TBC1D15 itself targets Rab7 at different cellular locations for distinct biological purposes. Endogenous TBC1D15 localizes primarily to mitochondrial membranes, where it participates in mitophagy and ER-curated mitochondrial fission (Onoue *et al*, 2013; Yamano *et al*, 2014; Wong *et al*, 2018). We now find that mitochondrial

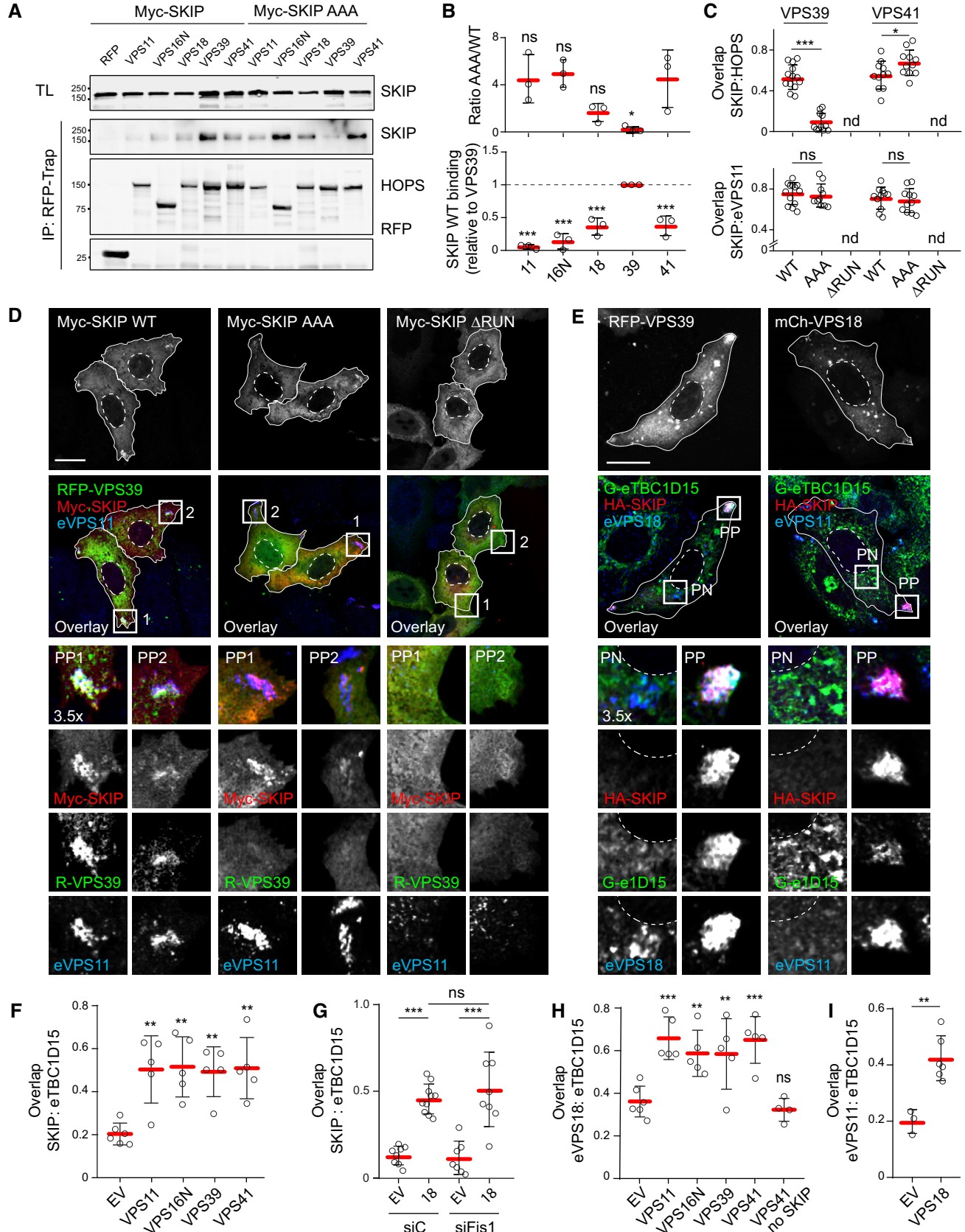

**Figure 9.**

**Figure 9.   Rab7 interactions influence SKIP-associated HOPS complex assembly.**

A–D   Effects of SKIP KMI motif on HOPS complex recruitment. (A) Analysis of SKIP/HOPS interactions by co-IP of RFP/mCherry-HOPS subunits with Myc-SKIP versus the KMI motif mutant AAA from HEK293T cells using RFP-trap beads. Representative immunoblots against RFP and Myc are shown, IP: immunoprecipitation, TL: total lysate. (B) Quantification of SKIP/HOPS interactions, *n* = 3 independent experiments. *Top graph*: fold change in binding of HOPS subunits to AAA mutant relative to wild type (WT) SKIP (1.0), significance: one-way ANOVA test (relative to VPS41). *Bottom graph*: Co-IP per HOPS subunit relative to VPS39, significance: one-way ANOVA test (relative to VPS39), *P < 0.05, ***P < 0.001, ns: not significant. (C) Colocalization (Mander's overlap) between SKIP and RFP-VPS39 (images shown in D) or mCherry-VPS41 (images shown in Fig EV4C), *n* ≥ 10 images (2 ≥ cells per image) analysed per condition from 2 independent experiments. (D) Representative confocal images of fixed HeLa cells ectopically expressing Myc-SKIP, KMI motif mutant (AAA) or RUN domain truncation mutant (ΔRUN) (*red*) together with RFP-VPS39 (*green*), immunolabelled for endogenous VPS11 (eVPS11, *blue*).

E–I   Effect of HOPS overexpression on recruitment of TBC1D15 to the SKIP compartment. (E) Representative confocal images of fixed HeLa cells harbouring endogenous GFP-TBC1D15 (Ge1D15, *green*) and ectopically expressing HA-SKIP (*red*) together with mCh-VPS18 or RFP-VPS39 (*white*), immunolabelled for endogenous VPS18 or VPS11 (*blue*), as indicated (see also Fig EV4D). (F–I) Colocalization of SKIP and HOPS complex with TBC1D15 as a function of HOPS subunit overexpression. (F, H) Mander's overlap between (F) SKIP or (H) eVPS18 and endogenous TBC1D15, *n* ≥ 5 images (2 ≥ cells per image) analysed per condition from 2 independent experiments, significance: one-way ANOVA test (relative to EV), **P < 0.01, ***P < 0.001, ns: not significant. (G, I) Mander's overlap between (G) SKIP or (I) eVPS11 and endogenous TBC1D15 as a function VPS18 overabundance, with or without FIS1 depletion (siFIS1), *n* ≥ 3 images (2 ≥ cells per image) analysed per condition from 2 or more independent experiments.

Data information: Cell and nuclear boundaries are demarcated with solid and dashed lines, respectively, and zoom insets (3.5×) highlight select peripheral (PP) and perinuclear (PN) cell regions, as indicated, all scale bars: 10 μm. Graphs report mean (red line) of sample values (open circles), and error bars reflect ± SD. Unless stated otherwise, significance was assessed using two-tailed Student's *t*-test: *P < 0.05, **P < 0.01, ***P < 0.001, ns: not significant, nd: not determined.
Source data are available online for this figure.

TBC1D15 appears to coexist in equilibrium with its cytosolic pool, and that Arl8b/SKIP/HOPS draws from the latter to conduct removal of Rab7 from select MVBs. In this way, SKIP connects TBC1D15-mediated Rab7-to-Arl8b conversion with plus-end-directed transport of MVBs into the cell periphery. Such coupling of membrane identity to directed motility echoes the famed Rab5-to-Rab7 conversion taking place earlier in the lifespan of an endosome when maturation is linked to transport towards the perinuclear proteolytic hub.

Spatiotemporal control of GTPase association with their target membranes bears consequences for transport of membranes and cargoes towards and away from relevant sub-compartments within the endolysosomal system. As the Rab5-to-Rab7 handover mechanism marks the timing of early-to-late transitions along the endocytic route, coupled to MVB formation and transport of these structures towards the cell's interior, the Rab7-to-Arl8b conversion described here defines a further membrane identity switch coupled to movement of select organelles back into the periphery. Arl8b has

**Figure 10.   TBC1D15 interacts with the HOPS complex to inactivate and remove Rab7 from membranes selected by SKIP.**

A, B   *In situ* SKIP/TBC1D15 complex formation assayed using proximity-based biotin ligation (BioID). (A) Neutravidin precipitates (PD) from biotin-treated HEK293T cells ectopically expressing GFP-TBC1D15 (GFP-1D15) together with HA-BioID-SKIP or HA-BioID-EV. Representative immunoblots against GFP, HA and VPS18 are shown, TL: total lysate. (B) Quantification of biotinylation of GFP-TBC1D15 (*top graph*) and endogenous VPS18 (eVPS18, *bottom graph*) by BioID-SKIP, expressed relative to BioID-EV (1.0), *n* = 4 independent experiments.

C   Analysis of TBC1D15/HOPS interactions by co-IP of RFP/mCherry-HOPS subunits with GFP-TBC1D15 from HEK293T cells using GFP-trap beads. *Left panel*: schematic of HOPS complex composition, with subunits tested marked in yellow. *Right panel*: representative immunoblots against GFP and RFP; IP: immunoprecipitation, TL: total lysate.

D–H   Effects of VPS18 depletion on the recruitment of TBC1D15 to the SKIP complex. (D) Representative confocal images of fixed HeLa cells harbouring endogenous GFP-TBC1D15 (G-e1D15, *green*), transfected with either control siRNA (siC) or oligo pool targeting VPS18 (siVPS18) and ectopically expressing HA-SKIP (*red*), immunolabelled against endogenous VPS18 (eVPS18, *blue*) and HA, scale bar: 10 μm. Cell and nuclear boundaries are demarcated with solid and dashed lines, respectively, and zoom insets (3.5×) highlight select peripheral (PP) and perinuclear (PN) cell regions. (E) Colocalization (Mander's overlap) between SKIP and endogenous TBC1D15 as a function of VPS18 depletion, *n*~siC~ = 9, *n*~siVPS18~ = 11 images (2 ≥ cells per image) analysed from 2 independent experiments. (F) Neutravidin precipitates (PD) from biotin-treated HeLa cells ectopically expressing GFP-TBC1D15 (GFP-1D15) together with BioID-SKIP or BioID-EV. Representative immunoblots against GFP, HA and VPS18 are shown, TL: total lysate. (G) Quantification of GFP-TBC1D15 biotinylation by BioID-SKIP over BioID-EV in response to VPS18 depletion expressed as % of control (siC), *n* = 4 independent experiments performed in HeLa or HEK293T cells. (H) Quantification of VPS18 protein abundance from experiments in G, *n* = 4.

I   Effect of TBC1D15 activity on Rab7 displacement from the SKIP compartment. Colocalization (Mander's overlap) between Rab7 and TBC1D15 (WT or R417A mutant) in the presence of either SKIP or PLEKHM1 combined with the indicated VPS proteins, *n* ≥ 5 images (2 ≥ cells per image) analysed from 2 independent experiments (see also Fig EV5D).

J   Proposed model. *Left panel*: active (*yellow star*) membrane-bound Rab7 can partner with its effector RILP to mediate dynein-dependent minus-end-directed transport of late endosomes and/or lysosomes into the perinuclear region, towards the microtubule organizing centre (MTOC). *Left middle panel*: a subpopulation of Rab7-positive vesicles acquires Arl8b, forming a hybrid identity compartment. Arl8b in turn recruits the effector SKIP through the N-terminal RUN domain (*red*), as well as VPS41 and the core HOPS subunits to endosomes. *Right middle panel*: SKIP is now in the position to engage Rab7 using its C-terminal segment featuring the KMI motif. This promotes acquisition of VPS39 and reconstitution of the HOPS complex, providing a platform for recruitment of the Rab7 GAP TBC1D15 (*yellow*) not associated with mitochondrial membranes. *Right panel*: the resulting Arl8b/SKIP/HOPS/TBC1D15 complex induces inactivation (*grey star*) and removal of Rab7 from the membrane, while maintaining Arl8b/SKIP identity and liberating the endosome for kinesin-dependent transport towards the microtubule plus-end, into the cell periphery. Regulated disengagement of Rab7 thus enables orderly selection of transport route.

Data information: Graphs report mean (red line) of sample values (open circles), and error bars reflect ± SD. All significance was assessed using two-tailed Student's *t*-test: *P < 0.05, ***P < 0.001, ns: not significant.
Source data are available online for this figure.

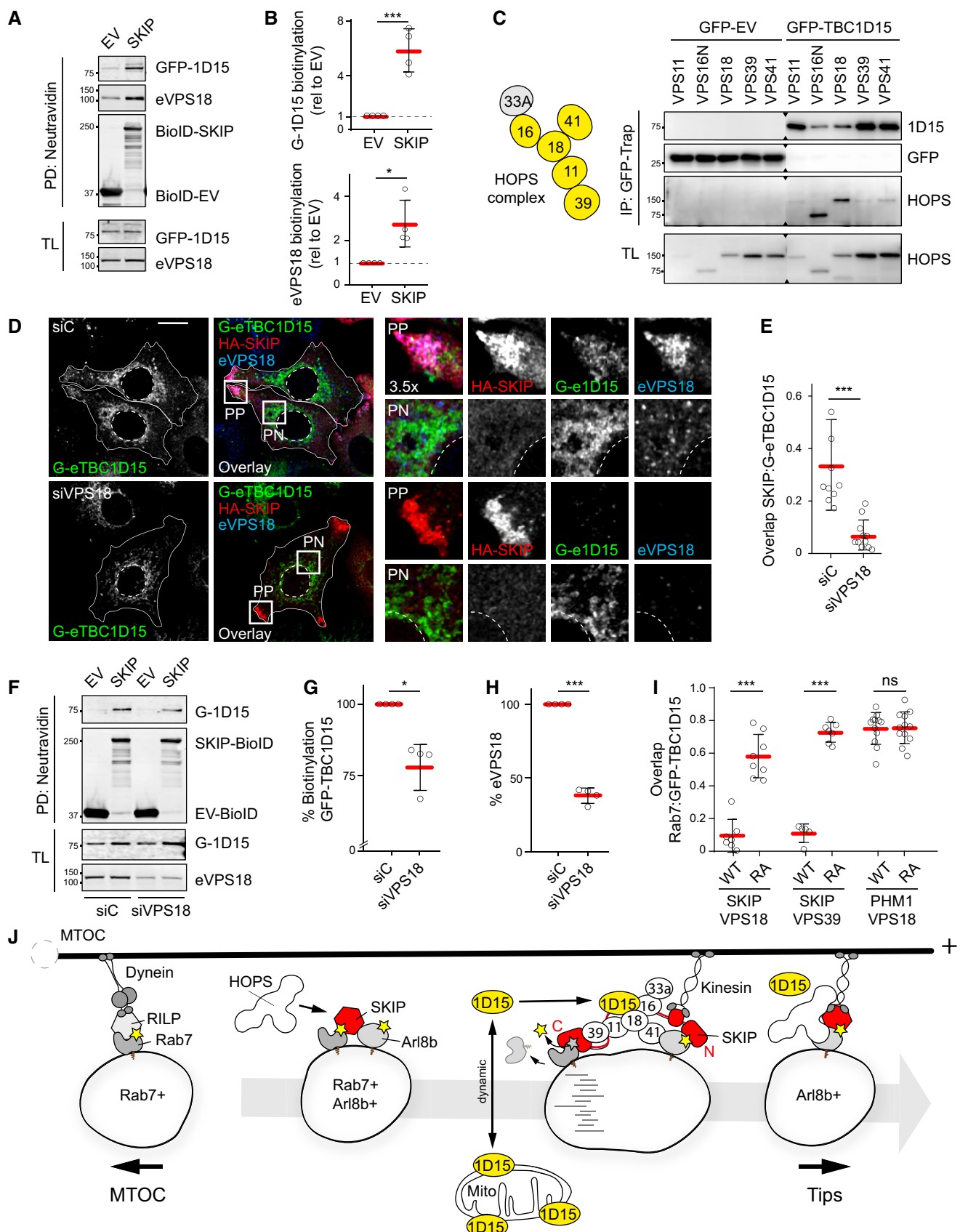

**Figure 10.**

been described to target later organellar profiles as compared to Rab7 (Hofmann & Munro, 2006; Garg *et al*, 2011). On the one hand, our description of SKIP-dependent ejection of Rab7 to form a *temporally* later Arl8b-positive single identity compartment support this order of events. Conversely, ultrastructural evaluation indicates that Rab7 together with RILP preside over *morphologically* "later" organelles, as compared to the peripheral MVBs targeted by Arl8b and SKIP in the same cells. Taken together with recent work by others (Johnson *et al*, 2016), our observations argue that the definition of what constitutes mature and/or late organelles within the cell's heterogeneous acidic and proteolytic repertoire is ever more complex and intriguing. Our findings further support the emerging paradigm of Rab7 as a central regulatory node in the endolysosomal system, capable of a wide range of partnerships necessary to manage the diverse lifestyles of acidic and proteolytic compartments.

# Materials and Methods

## Cell lines and culturing

HEK293T (human embryonic kidney) and HeLa (human cervical carcinoma) cell lines were purchased from ATCC and cultured in DMEM (Gibco) supplemented with 7.5% FCS. MelJuSo (human melanoma) cells were kindly provided by G. Riethmuller (LMU, Munich) and cultured in IMDM (Gibco) supplemented with 7.5% foetal calf serum (FCS, Greiner). HeLa cells harbouring endogenous CD63, Arl8b, or TBC1D15 N-terminally tagged with GFP were generated using CRISPR/Cas9. To create an endogenous GFP-tag vector, the CMV promotor from mGFP-C1 was replaced for MCS (linker) sequence and vector ATG site was mutated (see primer list). Custom primers (see list) for 5′ and 3′ genomic flanking regions for genes of interest were used to clone the gene-specific homology domains (HDR) into the vector near the mGFP (see primer list). If necessary, PAM sequences of the corresponding CRISPRs were mutated. HDR-mGFP and specific CRISPR constructs (pX330, see gRNA sequences below) were transfected into HeLa cells. GFP-positive cells were subsequently sorted, and colonies derived from single cells were expanded and validated. All cell lines were maintained at 37 degrees, 5% $CO_2$, and routinely (negatively) tested for mycoplasma contamination (Lonza, LT07-318).

| Primer name | Sequence |
|---|---|
| Linker MCS FW | TATGGATATCCTCGAGCGTACGAGCGCTAAG CTTATTTAAATGCGGCCGCACCGGTG |
| Linker MCS RV | CTAGCACCGGTGCGGCCGCATTTAAATAAGCTTA GCGCTCGTACGCTCGAGGATATCCA |
| Mut ATG endo tag FW | GCACCGGTGCTAGCGTGAGCAAGGGCGAGGAGCTG |
| Mut ATG endo tag RV | CTTGCTCACGCTAGCACCGGTGCGGCCGCATTTAAATAAG |
| CD63 Endo NdeI FW | CCCACACATATGCCGCAGCTGTTACCGCGTC |
| CD63 Endo NheI RV | CCCAGCTAGCCATGGCTGCCGGGGCCTGGGGCAG |

(continued)

| Primer name | Sequence |
|---|---|
| CD63 Endo BglII FW | CCCAAGATCTATGGCGGTGGAGGGAGGAAT |
| CD63 Endo EcoRI RV | CCCAGAATTCAAGAGAAAATCAGGTGAGGA |
| CD63 gRNA FW | CACCGCCGGCAGCCATGGCGGTGGA |
| CD63 gRNA RV | AAACTCCACCGCCATGGCTGCCGGC |
| Rab7GAP nEndo NdeI FW | CCCCACATATGGTCACAGAGCTAATAAGTAATGTG |
| Rab7GAP nEndo PinAI RV | CCCAACCGGTCATGTTTCCTGCGCGTGCCTGG |
| Rab7GAP nEndo BglII FW | CCAAGATCTGCGGCGGCGGGTGTTGTGAG |
| Rab7GAP nEndo EcoRI RV | CCCAGAATTCCACAAACGCTGTCAAATAATTC |
| Rab7GAP gRNA FW | CACCGACGCGCAGGAAACATGGCGG |
| Rab7GAP gRNA RV | AAACCCGCCATGTTTCCTGCGCGTC |
| ARL8b nEndo BglII FW | ACTACGAGATCTCTGGCGCTCATCTCCCGC |
| ARL8b nEndo EcoRI RV | ACTACGGAATTCCCGCCAACACGCGGGA |
| ARL8b nEndo NheI RV | ACTACGGCTAGCCATGATGGCGGTCGGGAGCG |
| ARL8b nEndo NotI FW | ACTACGGCGGCCGCCTGCGGAGCATGCCCACc |
| ARL8b gRNA FW | CACCGCTCCCGTCCGTTCTCGCTCC |
| ARL8b gRNA RV | AAACGGAGCGAGAACGGACGGGAGC |

## DNA constructs

GFP-Rab7 and Myc-Rab7 have been described before (Jordens *et al*, 2001), and GFP-Rab7 Q67L and T22N were gifts from P. Chavrier (Meresse *et al*, 1995). mCherry-Rab7 [G. Voeltz, Addgene plasmid #61804 (Rowland *et al*, 2014)] was used to generate mCherry-Rab7 Q67L and T22N mutants by site-directed mutagenesis using standard protocols. Arl8b-GFP was a gift from S. Munro (Hofmann & Munro, 2006) and was used to generate Myc-Arl8b by cloning it into the 2xMyc-C1 vector (KpnI/BamHI restriction sites). GST-Rab7 and GST-Arl8b were cloned into a modified pGEX-GST bacterial expression vector at KpnI/XhoI restriction sites from canine Rab7a cDNA (NM_001003316.1) and murine Arl8b cDNA (NM_026011.3), respectively, using a ligation-independent cloning approach (Luna-Vargas *et al*, 2011). Canine Rab5 cDNA (coding for aa 1–210, NM_001003317.2) was coned into pRP-261 bacterial expression vector BamHI/EcoRI restriction sites. Human Rab9a cDNA (NM_004251) was cloned into pET15B-His bacterial expression vector. GST-Ran has been previously described (Izaurralde *et al*, 1997).

Human SKIP cDNA (Rosa-Ferreira & Munro, 2011) was cloned into mRFP-C1 (BglII/HindIII) and 2xMyc-C1 (BglII/EcoRI) vectors using, resulting in RFP-SKIP and Myc-SKIP, respectively. HA-ER-GFP-SKIP was created by substituting YFP for a linker-HA fragment (SalI/NheI restriction sites) and CFP for GFP (BglII/BamHI restriction sites) from the original pcDNA3 YFP-ER-CFP (Michalides *et al*,

2004). SKIP was then cloned into the resulting vector (BglII/NotI restriction sites). HA-TurboID-SKIP and the corresponding vector control were generated as follows. 2xHA-TurboID fragment was amplified from 3xHA-TurboID-NLS-pcDNA3 (A. Ting Addgene plasmid #107171) and cloned into mGFP-C1 vector (NheI/BglII restriction sites) to substitute GFP, resulting in 2xHA-TurboID-C1. Subsequently, SKIP was cloned into 2xHA-TurboID-C1 at BglII/EcoRI restriction sites.

Myc-SKIP truncations and RFP-SKIP KMI mutants were generated using *In Vivo* Assembly (IVA) method (Garcia-Nafria *et al*, 2016). In short, a reaction mix containing 0.2 μM forward and 0.2 μM reverse primer (18 nucleotides specifically designed), 200 μM dNTPs, 0.25 ng template, 1× Phusion HF Buffer, 0.25 μl Phusion DNA Polymerase, and filled to 25 μl with MilliQ was amplified using the following programme 98°C 3 min; (98°C 10 s; 60°C 30 s; 72°C 15 s/kb) × 25 cycles; 72°C 5 min; 4°C forever. Reaction products were digested with 1 μl DpnI for 15 min at 37°C and 2 μl subsequently transformed into the *Escherichia coli* DH5α strain. All constructs used in the study were sequence verified.

Human RILP cDNA (Jordens *et al*, 2001) was cloned into mRFP-C1 and HA-C1 vectors using BglII and HindIII restriction enzymes, PLEKHM1-FLAG (Wijdeven *et al*, 2016). GFP-FYCO1 was a gift from T. Johansen (Pankiv *et al*, 2010).

Murine His-TBC1D15, a gift from X. Zhang (Zhang *et al*, 2005), was used to clone TBC1D15 into mGFP-C1 and mRFP-C1 (NcoI/SpeI restriction sites). GFP-TBC1D15 was then used to create the inactive mutant R417A and siRNA-resistant mutants by site-directed mutagenesis according to standard protocols. GFP-TBC1D5 was a gift from M. Seaman (Seaman *et al*, 2009).

HOPS complex constructs for mCherry-VPS11, mCherry-VPS16N, mCherry-VPS18, RFP-VPS33B, RFP-VPS39 and mCherry-VPS41 have been previously described (van der Kant *et al*, 2015).

## DNA and siRNA transfections

DNA transfections of HeLa and MelJuSo cells were carried out using Effectene (Qiagen #301427), according to manufacturer's protocols. HEK293T cells were transfected using Polyethylenimine (PEI) (Polysciences Inc. #23966) as follows: medium without supplements was mixed with DNA and PEI (ratio 1:3), incubated at RT for 30 min and added to the cells in droplets. PEI-transfected cells were cultured for 18–24 h prior to further analysis.

For siRNA transfections, oligos targeting TBC1D2 (siGenome #020463), TBC1D5 (siGenome #020775), TBC1D15 (siGenome #016209), TBC1D17 (siGenome #014409), RILP (siGenome #008787), SKIP (siGenome #022168), FIS1 (siGenome #020907) or VPS18 (siGenome #013178) were purchased from Dharmacon, as was the non-targeting siRNA (siCTRL #D-001206-13-20) used as the negative control throughout the study. Gene silencing was performed as follows (for 24-well plate format): 50 μl siRNA diluted to 500 nM in RNA buffer (Dharmacon #B-002-000-UB-100) was mixed with 1 μl DharmaFECT 1 (Dharmacon #T-2001-03) diluted in 49 μl supplement free medium, and reaction mixtures were incubated at RT for 20 min with continuous shaking. Subsequently, 400 μl of cells suspended in complete medium at 37.5 × 10³/ml was added to the transfection mixture and glass coverslips were added to samples intended for fixation and subsequent immunofluorescence imaging. For live cell imaging, the reaction was scaled up 4×, and cells were cultured on 35-mm glass bottom dishes (1.5 coverglass, MatTek). In all cases, following transfection, cells were cultured for 3 days prior to further analysis.

## RNA isolation, cDNA synthesis and qPCR

RNA isolation (Bioline), cDNA synthesis and quantitative RT–PCR (Roche) were performed according to the manufacturer's instructions. Obtained signal was normalized to GAPDH, and expression was calculated using the Pfaffl formula. The following primers were used for detection:

| TBC1D5 FW | 5′-GCTAGACCACAAGATTTAGGGC-3′ |
|---|---|
| TBC1D5 RV | 5′-CGCACCCACCTTAACCCATA-3′ |
| TBC1D15 FW | 5′-TGGAAAGACCAATGACCAAGAC-3′ |
| TBC1D15 RV | 5′-TCCACTATTACTTCGGCATCCT-3′ |
| TBC1D17 FW | 5′-CCCCGGCTATGAACCTGAC-3′ |
| TBC1D17 RV | 5′-GGCGGATGGACTTTAGCTCC-3′ |
| TBC1D2 FW | 5′-GAGGCGGTCCCCAAGAAAC-3′ |
| TBC1D2 RV | 5′-TTTCGTCGTAGAAGAACCAGC-3′ |
| Fis1 FW | 5′-GATGACATCCGTAAAGGCATCG-3′ |
| Fis1 RV | 5′-AGAAGACGTAATCCCGCTGTT-3′ |
| GAPDH FW | 5′-TGTTGCCATCAATGACCCCTT-3′ |
| GAPDH RV | 5′-CTCCACGACGTACTCAGCG-3′ |

## Antibodies, reagents and fluorescent dyes

### Confocal microscopy

The following primary antibodies were used for immunofluorescence: rabbit anti-Rab7 (Cell Signalling #D95F2), anti-MHC-II (Neefjes *et al*, 1990), anti-VPS11 (Abcam #ab125083), anti-VPS18 (Abcam #ab178416) and anti-TBC1D15 (Sigma # HPA013388), as well as mouse anti-CD63 NKI-C3 (Vennegoor & Rumke, 1986), anti-LAMP1 (H4A3, Santa Cruz), anti-TOM20 (Abcam #ab56783), anti-Myc (9E10, Sigma) and anti-FLAG M2 (Sigma #1804), as well as rat anti-HA (3F10, Roche). Secondary donkey anti-rabbit/anti-mouse (Invitrogen) or anti-rat (Biotium) Alexa-coupled antibodies were subsequently used for fluorescence detection, as applicable. Mitotracker Red CMXRos (Invitrogen) and SiR-Lysosome (Spirochrome #CY-SC012) were used to visualize mitochondria (100 nM, for 30 min) and late endosomes/lysosomes (2 μM, for 30–60 min), respectively. 4-OH-Tamoxifen (used at 0.1 μM final concentration) was a gift from W. Zwart (NKI).

### Western blotting

The following primary antibodies were used for detection by immunoblot: rabbit anti-mGFP (Rocha *et al*, 2009), anti-mRFP (Rocha *et al*, 2009) and anti-VPS18 (Abcam #ab178416), as well as mouse anti-RFP (5F8, Chromotek), anti-HA (HA.11 (16B12), Covance), anti-Myc (9E10, Sigma), anti-GST (2F3) (van Zeijl *et al*, 2007), anti-β-actin (AC-15, Sigma), anti-FLAG M2-PO (Sigma #A8592) and anti-HA-PO (Roche #12013819001). The following secondary detection agents were used when applicable: rabbit anti-mouse-PO (Dako # P0161), HRP-Protein A (Invitrogen #10-1023), IRDye 800CW goat anti-rabbit IgG (H+L) (Li-COR #926-32211), IRDye 800CW goat anti-

mouse IgG (H+L) (Li-COR #926-32210), IRDye 680LT goat anti-rabbit IgG (H+L) (Li-COR #926-68021) and IRDye 680LT goat anti-mouse IgG (H+L) (Li-COR #926-68020).

### Immunoprecipitation

Rabbit anti-TBC1D15 (Abcam #ab121396) was used to isolate TBC1D15 from lysates, as well as for subsequent immunoblotting.

### Immunogold EM

Biotinylated goat anti-GFP (#600-106-215) and anti-biotin (#100-4198) were both obtained from Rockland Immunochemicals, and anti-RFP (#R10367) was purchased from Molecular Probes.

### Fluorescence microscopy

Fixed sample preparation for confocal microscopy was performed according to standard protocols. Samples were mounted using ProLong Gold antifade (P10144, Invitrogen) and imaged on Leica SP8 microscope equipped with appropriate filters for fluorescence detection. Images were acquired using a Hcx PL 63 × 1.32 oil objective and 1–4 digital zoom as applicable. Alexa-405 was excited at $\lambda = 405$ nm and detected at $\lambda = 416$–470 nm; Alexa-488 was excited at $\lambda = 488$ nm and detected at $\lambda = 500$–550 nm; Alexa-568 was excited at $\lambda = 561$ nm and detected at $\lambda = 570$–621 nm; Alexa-647 was excited at $\lambda = 633$ nm and detected at $\lambda = 642$–742 nm. Pixel plot graphs were created by using the line profile tool of LAS-AF. Colocalization was reported as Mander's overlap calculated using JACoP plug-in for ImageJ (displayed as protein A: protein B = amount of protein A overlapping with protein B).

For live cell imaging, cells were grown on 35-mm glass bottom dishes (1.5 coverglass, MatTek) coated with poly-d-lysine. Cells were treated as applicable, and time-lapses were collected on the Dragonfly spinning disc microscope or Leica WLL microscope, both adapted with climate control chambers. Images were acquired using Leica Hcx PL 100 × 1.30 or 63 × 1.32 oil objectives, and data were analysed using Fiji software. Vesicle tracking was performed on the indicated channels across 100 or 50 consecutively collected frames using TrackMate for Fiji (vesicle diameter = 1 μm; thresholds and other parameters were chosen as appropriate based on control samples; the same parameters were used across all samples and experiments within a given group).

Fractional distances were calculated as follows. Fluorescence intensities along multiple line ROIs (assessed by using the line profile tool of LAS-AF) were normalized to median, and background pixels were excluded from the analysis by determining the signal threshold. Distances corresponding to the remaining (vesicular) pixels were plotted as fractions of the distance from the nucleus to the plasma membrane along the same line.

### Fluorescence recovery after photobleaching

Fluorescence recovery after photobleaching (FRAP) experiments on Rab7 were performed using MelJuSo cells stably expressing ectopic GFP-Rab7, and those on TBC1D15 were performed on endogenous GFP-TBC1D15 in HeLa cells. Leica's LAS-X FRAP wizard software in combination with a Leica SP8 WLL or SP5 microscope equipped with a climate chamber was used for image acquisition. A pre-bleach image was collected using 10% laser power. Subsequently,

photobleaching was performed on a single endosome or mitochondria-containing spots for 0.75 s at 100% laser power, resulting in a 90–100% reduction of the fluorescence signal. Sequential images were then collected at 10% laser power at 5-s or 0.4-s intervals for 20–60 frames. Image analysis was performed using ImageJ. Maximum fluorescence recovery (i.e. fluorescence prior to bleaching) was set at 100%, while post-bleach values were corrected for background and bleaching effects (Reits & Neefjes, 2001).

### Correlative light and electron microscopy (CLEM) of lysosomal distribution and ultrastructure

HeLa cells expressing endogenous CD63-GFP were grown to a density of $3 \times 10^5$ cells per coverslip. Cells were incubated with SiR-Lysosome (Spirochrome) in complete DMEM for 30 min before fixation in $1 \times$ PHEM buffer containing 4% PFA and 0.1% glutaraldehyde. Fluorescence imaging of fixed cells was performed using a Deltavision RT widefield microscope system (GE Healthcare) equipped with a 100×/1.4NA oil immersion lens and a Cascade II EM-CCD camera (Photometrics). Multi-channel $Z$-stacks were captured and deconvolved using Softworx 6.5.2 (GE Healthcare). After fluorescence imaging, cells were flat embedded for electron microscopy using established protocols as previously described (Oorschot et al, 2002). Cells were postfixed with 1% $OsO_4$ with 1.5% $K4Fe(II)(CN)6$ in $1 \times$ PHEM for 1 h on ice, followed by washing steps in $ddH2O$ and stained with 2% uranyl acetate in $ddH_2O$ at room temperature, followed by further washing steps with $ddH2O$. Finally, samples were subjected to a graded ethanol series for dehydration. After dehydration, samples were embedded in Epon resin (ratio: 12 g Glycid Ether 100, 8 g dodecenylsuccinic anhydride, 5.5 g methylnadic anhydride, 560 μl N-benzyldimethylamine). After Epon polymerization, the resin blocks were removed from the coverslips and prepared for EM as reported before with slight modifications (Fermie et al, 2018). Regions of interest selected based on fluorescence imaging were cut out as blocks, from which serial sections (70 nm or 300 nm) were made with an ultramicrotome (Leica) and collected on carbon coated copper TEM grids. Thin sections were imaged using a Tecnai T12 TEM (Thermo Scientific), equipped with Veleta (EMSIS GmbH). Stitched EM images were collected by SerialEM: https://www.cambridge.org/core/journals/microscopy-and-microanalysis/article/serialem-a-program-for-automated-tilt-series-acquisition-on-tecnai-microscopes-using-prediction-of-specimen-position/DB5EA0250C3803C2D7C145CD13B4ADE6 and processed by IMOD software. Tomograms from 300 nm sections were made using a Tecnai 20 TEM (Thermo Fisher Scientific) operating at 200 kV, equipped with an Eagle 4K × 4K CCD camera running Xplore3D (Thermo Fisher Scientific) software. Single tilt image series were automatically collected with 1° tilt increments from −60° to +60°. Tomographic reconstructions were generated using IMOD software (Mastronarde & Held, 2017).

### Immunogold EM

HeLa cells transfected as indicated were prepared for cryosectioning, as described (Peters et al, 2006; van Elsland & van Kasteren, 2016). Briefly, cells were fixed for 24 h in freshly prepared 2% paraformaldehyde in 0.1 M phosphate buffer. Fixed cells were scraped, embedded in 12% gelatin (type A, bloom 300, Sigma) and

cut with a razor blade into 0.5-mm$^3$ cubes. The sample blocks were infiltrated in phosphate buffer containing 2.3 M sucrose. Sucrose-infiltrated sample blocks were mounted on aluminium pins and plunged in liquid nitrogen. The vitrified samples were stored under liquid nitrogen. Ultrathin cell sections of 75 nm were obtained essentially as described elsewhere (van Elsland & van Kasteren, 2016). Briefly, the frozen sample was mounted in a cryo-ultramicrotome (Leica). The sample was trimmed to yield a squared block with a front face of about 400 × 300 μm (Diatome trimming tool). Using a diamond knife (Diatome) and antistatic devise (Leica), a ribbon of 75-nm-thick sections was produced that was retrieved from the cryo-chamber with the lift-up hinge method (Bos et al, 2011). A droplet of 2.3 M sucrose was used for section retrieval. Obtained sections were transferred to a specimen grid previously coated with formvar and carbon. Grids containing thawed cryo sections of cells fixed with 2% paraformaldehyde were incubated on the surface of 2% gelatin at 37°C. Subsequently, sections were rinsed to remove the gelatin and sucrose, blocked with 1% BSA in PBS and then double-labelled with the indicated antibodies and 10-nm and 15-nm protein A-coated gold particles (CMC, Utrecht University) (Slot et al, 1991). Next, grids were embedded in 1.8% methylcellulose and 0.6% uranyl acetate. EM imaging was performed with an Tecnai 20 transmission electron microscope (FEI) operated at 120 kV acceleration voltage.

### Protein expression and purification

Recombinant protein expression of GST-Rab7, GST-Arl8b and GST-Rab5 was performed in the *E. coli* Rosetta2 (DE3) strain. Briefly, transformed cultures were inoculated in LB at 37°C till OD$_{600}$ 0.6 and expression was induced with IPTG at 16°C overnight. Cultures were harvested in 50 mM HEPES pH 7.5, 250 mM NaCl, 1 mM EDTA and 1 mM DTT, freshly supplemented with complete EDTA-free protease inhibitors (Roche Diagnostics) and lysed by microtip sonication. For purification of GST-fusion proteins, lysates were incubated with Glutathione Sepharose® 4B affinity resin (GE Healthcare). Recombinant proteins were eluted with glutathione, and eluates were purified by size-exclusion chromatography using a Superdex® 75 (16/600) gel filtration column (GE Healthcare) equilibrated with buffer 50 mM Tris–HCl pH 8.0, 150 mM NaCl. For purification of His-Rab9, lysates were incubated with preequilibrated Talon Co$^{2+}$ resin (Clontech). The resin was then washed extensively with 20 mM HEPES pH 7.5, 200 mM NaCl and 8 mM β-mercaptoethanol, and His-Rab9 was eluted in washing buffer supplemented with 500 mM imidazole. His-TBC1D15 was purified from HEK293T cells lysed in lysis buffer (0.5% Triton X-100, 20 mM HEPES pH 7.5, 200 mM NaCl, 8 mM β-mercaptoethanol) freshly supplemented with complete EDTA-free protease inhibitors. Clarified supernatant was incubated with preequilibrated Talon Co$^{2+}$ resin (Clontech), and the same washing and elution steps were followed as for His-Rab9 above.

Purity, apparent molecular weight and concentration of recombinant proteins were evaluated by SDS–PAGE followed by InstantBlue or Coomassie staining.

### In vitro γ$^{32}$P-GTP loading assays

GTPase assays were performed as previously described (Askjaer et al, 1999). Briefly, GST-Rab7, GST-Rab5, His-Rab9 and GST-Ran were produced and purified as described in "Protein Expression and Purification".

GTPases were loaded with [γ-$^{32}$P]GTP (10 mCi/ml, > 5,000 Ci/mmol; GE Healthcare) in the presence of 10 mM EDTA. Subsequently, MgCl$_2$ was added to a final concentration of 20 mM, followed by gel filtration on a Bio-Spin 6 column (Bio-Rad) equilibrated with buffer (0.1 M Tris–HCl, pH 7.5, 10 mM MgCl$_2$, 2 mM DTT, 0.5 M NaCl) to remove free [γ-$^{32}$P]GTP. Reaction mixtures containing GST-fusion proteins in reaction buffer (40 mM Tris–HCl, pH 8.0, 50 mM NaCl, 8 mM MgCl$_2$, 1 mM DTT, 0.5 mM non-radioactive GTP, 0.1 mg/ml BSA, 1 mM Na$_3$PO$_4$, 1% (v/v) glycerol) were assembled on ice and incubated with increasing concentrations of His-TBC1D15 for 15 min. Reactions were stopped by adding 1 ml of charcoal suspension (7% w/v charcoal, 10% v/v ethanol, 0.1 M HCl, 10 mM KH$_2$PO$_4$), and the mixture was centrifuged for 5 min. Release of [$^{32}$P]-orthophosphate was quantified by liquid scintillation counting of the supernatant.

### Co-immunoprecipitation

HEK293T cells were lysed for 30 min in lysis buffer (0.8% NP-40, 50 mM NaCl, 50 mM Tris–HCl pH 8.0, 5 mM MgCl$_2$) freshly supplemented with complete EDTA-free protease inhibitors (Roche Diagnostics). Supernatants obtained following centrifugation (10 min at 12,000 × g) were incubated with GFP-Trap_A, RFP-Trap_A or Myc-Trap_A agarose (Chromotek) rotating at 4°C for 1 h. Beads were then washed 4× in lysis buffer, ensuring that buffer was completely removed after the last wash using a needle before the addition of SDS Sample Buffer (containing 5% β-mercaptoethanol), followed by 5-min incubation at 95°C. Co-immunoprecipitated proteins were separated by SDS–PAGE for Western blotting as indicated.

### Proximity-based labelling (BioID)

Proximity-based labelling experiments were performed as described before (Sapmaz et al, 2019). Briefly, HEK293T or HeLa cells were transfected with either 2xHA-TurboID vector control or 2xHA-TurboID SKIP and GFP-TBC1D15 (48 h following siRNA transfection, if applicable). After 24 h, the cells were treated with biotin (Sigma, B4639) at 50 μM final concentration for 5–6 h. Cells were lysed for 20 min in lysis buffer (0.8% NP-40, 150 mM NaCl, 50 mM Tris–HCl (pH 8.0), 5 mM MgCl$_2$) freshly supplemented with protease inhibitors (Roche). SDS (1%) was then added to the supernatant following centrifugation (20 min, 4°C, 20,817 × g). High-capacity neutravidin beads (Thermo Scientific, 29202) were then added and samples were incubated rotating at 4°C overnight. Samples were washed 4× in buffer (0.8% NP-40, 150 mM NaCl, 50 mM Tris–HCl pH 8.0, 1% SDS and 5 mM MgCl$_2$) and eluted with SDS Sample Buffer (containing 5% B-mercaptoethanol), followed by 5-min incubation at 95°C. Biotinylated proteins were separated by SDS–PAGE and detected by Western blotting.

### SDS–PAGE and immunoblotting

Samples were separated by 8% or 10% acrylamide gel and stained using InstandBlue (Expedeon) or transferred to Nitrocellulose (0.45 μm, GE Healthcare) or Immobilon-P PVDF membrane (0.45 μm, Millipore) at 300 mA for 3 h. Membranes were blocked

in 5% milk (skim milk powder, LP0031, Oxiod) in 1× PBS (P1379, Sigma-Aldrich), incubated with primary antibodies diluted in 5% milk/0.1% PBS-Tween-20 (PBST) for 1 h, washed 3 × 10 min in 0.1% PBST, incubated with the secondary antibody diluted in 5% milk/0.1% PBST for 30 min and washed 3× in 0.1% PBST. Antibody signals on Nitrocellulose membranes were detected using Odyssey CLx (Li-COR), and those on PVDF membranes were collected on an Amersham Imager 600 (GE Healthcare).

## Statistics

All error bars correspond to SD of the mean. Statistical evaluations report on Student's test (analysis of two groups) or one-way ANOVA analyses (analysis of three or more groups), as described in the corresponding figure legends, *P < 0.05, **P < 0.01, ***P < 0.001, ns: not significant and nd: not determined.

**Expanded View** for this article is available online.

## Acknowledgements

We thank S. Munro (MRC) for the generous gift of Arl8b-GFP and SKIP cDNA, as well as M. Seaman (CIMR) and T. Johansen (IMB) for kindly providing GFP-TBC1D5 and GFP-FYCO1 constructs, respectively. We thank R. Kim of the LUMC protein facility for help with secondary structure prediction in designing the SKIP truncation mutants, B. Koster for access to the LUMC electron microscopy facility, as well as L. Voortman and A. van der Laan-Boonzaier of the LUMC and L. Oomen and L. Brocks of the NKI for fluorescence microscopy facility support. We acknowledge technical assistance of the Cell Microscopy Core at the UMC Utrecht, especially C. de Heus. This work was supported by the NWO TOP and ERC Adv grants awarded to J. Neefjes.

## Author contributions

MLMJ, JB and IB designed the study and performed and analysed the majority of experiments. BC performed *in vitro* interaction assays, NL and JF carried out CLEM studies, DE performed immunoEM, RHW identified the KMI motif in SKIP, and CK performed GTPase assays. JLLA, SYZ, LJ and DH provided technical support. JK and RK advised on the project. IB, JB and MLMJ wrote the manuscript with input from JN. JN and IB jointly coordinated the study.

## Conflict of interest

The authors declare that they have no conflict of interest.

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
