## [Review Process File · The EMBO Journal]

The SKIP-HOPS complex recruits TBC1D15 for a Rab7-to-Arl8b identity switch controlling transport of late endosome

Marlieke Jongsma, Jeroen Bakker, Birol Cabukusta, Nalan Liv, Daphne van Elsland, Job Fermie, Jimmy Akkermans, Coenraad Kuijl, Sabina van der Zanden, Lennert Janssen, Denise Hoogzaad, Rikvan der Kant, Ruud Wijdeven, Judith Klumperman, Ilana Berlin, and Jacques Neefjes

Review timeline:	Submission date:	23rd Apr 19
	Editorial Decision:	29th May 19
	Revision received:	20th Nov 19
	Editorial Decision:	18th Dec 19
	Revision received:	10th Jan 20
	Accepted:	17th Jan 20

Editor: Elisabetta Argenzio

Transaction Report:

1st Editorial Decision

29th May 19

Thank you for submitting your manuscript entitled "SKIP-HOPS recruits TBC1D15 for a Rab7-to-Arl8b identity switch to control late endosome transport" to The EMBO Journal. Please accept my apologies for the lengthy review process due to a belated report. Your study has been sent to three reviewers for evaluation, whose reports are enclosed below.

As you can see, the referees consider the work potentially interesting. However, they also raise several criticisms that need to be addressed before they can support publication in The EMBO Journal.

In particular, referee #1 stresses that TBC1D15 localization on endosomes relies solely on overexpression experiments. Thus, s/he requests you to assess the endosomal localization of endogenous TBC1D15 and, if this is the case, to prove that this is not an artifact of SKIP overexpression. In addition, this referee asks you to i) further characterize the interaction between TBC1D15 and HOPS, ii) perform rescue experiments with TBC1D15, iii) test the role of the other two Rab7-specific GAPs, and iv) determine the activity of endogenous Rab7 biochemically (e.g. by using GST-RILP probe). Along the same lines, Referee #2 finds that the roles of SKIP and HOPS in TBC1D15 recruitment need to be better clarified and requests you to show that Arl8B, Rab7, SKIP, HOPS complex and TBC1D15 are in the same complex. Referee #3 asks you to investigate the direct association of SKIP with GDP- versus GTP-Rab7 and to analyze the switch from Rab7-positive to Arl8b-positive endosomes in live cells.

Given the overall interest of your study, I would like to invite you to submit a revised version that addresses the above-mentioned points. Note that solving these issues as suggested by the referees is essential to warrant publication of your manuscript in The EMBO Journal. I should also add that it is our policy to allow only a single round of revision. Therefore, acceptance of your manuscript will depend on the completeness of your responses in this revised version.

REFEREE REPORTS

Referee #1:

In the present manuscript, Bakker et al. first investigate the spatial and dynamic relationship between the small GTPases RAB7 and ARL8b. These GTPases mediate the directional movement of late endosomes/lysosomes in an opposing fashion through engagement of their effectors RILP and SKIP, respectively. The balance between RAB7-RILP and ARL8b and SKIP activity this determines late endosome localization and also determines their functionality. In their overexpression system, ARL8b and SKIP drive LE/Lys towards the cell periphery, concomitant with a removal of RAB7 from these structures. They present evidence that the RAB7 GAP TBC1D15 can shuttle from mitochondria to peripheral late endosomes through the action of SKIP and the HOPS complex, which removes RAB7 from these endosomes thereby promoting correct LE/ly mobility. Thus, TBC1D15 regulates RAB7 activity, late endosome positioning and also functionality.

The data is of high quality and largely supportive of the authors' hypothesis. However, I do have serious concerns about the validity of the data as very important data are missing in the manuscript: As far as I can tell, all the data on localization of GFP-TBC1D15 (endogenously tagged) has been obtained from cells overexpressing SKIP and or ARL8b. The authors need to obtain evidence that TBC1D15 is present on endosomal and or lysosomal membranes in parental cells which do not overexpress anything to support their thesis. TBC1D15 strongly localizes to mitochondria and at least some of the data that the authors present (loss of lysosomal motility, for example) could also be due to its role in the recently discovered lysosome-mitochondria contacts. Depletion of TBC1D15 should strengthen these contacts so that enhanced lysosome-mitochondria tethering then perturbs lysosomal mobility and positioning. The TBC1D15 antibody from the human proteome atlas (HPA013388, probably the same antibody as the one used in this study) works very well to detect endogenous TBC1D15 in fixed cells. If TBC1D15 does indeed play a role on lysosomes, the authors should be able to detect it there. GFP-TBC1D15 should also localize to lysosomes under native conditions. Depletion of SKIP or HOPS should then result in lack of TBC1D15 recruitment to lysosomes. As it is, it is possible that overexpression of SKIP induces artificial recruitment of TBC1D15 to lysosomes. This is particularly relevant since HOPS components (VPS39) have been detected in lysosome-mitochondria contacts in yeast. Conceivably, SKIP/HOPS and TBC1D15 organize lysosome-mitochondria contacts and overexpression of SKIP results in a mislocalization of this machinery to lysosomes. The authors should also knock down FIS1, which will disrupt the mitochondrial localization of TBC1D15 and analyze RAB7 activity, lysosome positioning and the RAB7/ARL8b relationship. This could potentially serve to separately analyze the role of TBC1D15 on mitochondria and on lysosomes.

Also, the data on the TBC1D15-HOPS interaction is too preliminary for a quality journal like EMBOJ. The interaction is a core element of the authors' thesis and needs to be demonstrated more thoroughly (also see major points below). Lastly, I was a bit surprised that the authors only looked at TBC1D15 and ignored the other two RAB7 GAP proteins (also see below). In the major points listed below, I have summarized these and other concerns. Overall, the study is interesting and I am supportive of publication if (and only if) the authors can convincingly demonstrate the presence of TBC1D15 on endosomal membranes in the absence of overexpressed interaction partners.

Major points:

1. The authors should perform rescue experiments with TBC1D15. I realize that they have used multiple siRNAs but in the age of CRISPR/Cas9 they should also quickly knock out TBC1D15 and re-express it at low levels (lentiviral, preferably) to revert the resultant phenotype. They could also use a siRNA resistant cDNA. Maybe the authors tried this already and had problems with mis-localized, overexpressed TBC1D15? This should be stated as it would explain the somewhat outdated approach to use so many siRNAs.
2. The authors speculate that the control of RAB7 activity is required for efficient segregation of SKIP and RILP compartments. They then test the effect of TBC1D15 depletion. Given that TBC1D15 was reported to be localized to mitochondria, whereas two other RAB7 GAP proteins

were reported to act on late endosomes, this appears to be a somewhat odd initial choice? Several independent reports have shown that TBC1D2/Arms regulates RAB7 activity on endosomes. Jimenez-Orgaz et al, 2018, EMBOJ and Seaman et al., 2018, JCS have recently shown that TBC1D5 acts through retromer and controls RAB7 activity in a major way. How are these mechanisms related? I would think that depletion of TBC1D2 and TBC1D5 should result in similar effects on RAB7 and ARL8b compartments? According to Jimenez-Orgaz et al (2018), and Seaman et al. (2018), RAB7 becomes grossly hyperactivated on LE/Lys in the absence of retromer/TBC1D5. Is the SKIP/HOPS/TBC1D15 mechanism simply a parallel pathway to retromer/TBC1D5? The authors should also cite these two studies in their discussion and maybe discuss whether these are related /redundant/parallel to their newly discussed pathway.

3. Mitochondrial TBC1D15: As reported previously, the authors find endogenous TBC1D15 on mitochondria. What happens to SKIP/ARL8b or RAB7 when FIS1, the mitochondrial tether for TBC1D15, is depleted? FIS1 knockdown/KO results in a cytosolic dispersal of TBC1D15, which obviously greatly increases the free pool of TBC1D15. If the authors' thesis is correct, this should lead to changes in late endosome positioning. They could also use FIS1 depletion as a tool to separately analyze the function of TBC1D15 on lysosomes and on mitochondria.

4. Figure 4A: The authors use FRAP to investigate RAB7 membrane cycling as the only (indirect) tool to analyze the activity status/mobility of RAB7. Ideally, the authors would determine the nucleotide status of RAB7 directly. However, I realize this is not necessarily trivial so the authors could also use established effector assays such as the assay based on GST-RILP developed by Bucci and colleagues (PMID:19347311). Given that depletion of TBC1D15 results in a striking loss of RAB7 mobility, indicative of an arrest in the active state, there should be much more activated RAB7 binding to the GST-RILP probe? How does the effect compare to loss of TBC1D2 and TBC1D5?

5. Figure 1: The data on RILP and SKIP overexpression is quite clear. Yet, the authors could easily strengthen their conclusions by knocking down (or knockouts) endogenous SKIP and RILP. This, if the thesis is accurate, should lead to the opposite effects compared to overexpression.

6. Figure 7B: I do not like that both tested interactors are overexpressed, I have seen far too many unspecific results when large amounts of two proteins are overexpressed in HEK293 cells. Given that an interaction between TBC1D15 and HOPS is a crucial part of their story, the presented IP data is too preliminary. The authors should test whether the RFP-tagged HOPS components precipitate endogenous TBC1D15. In addition, they also need to show that their endogenously GFP-tagged TBC1D15 precipitates the endogenous HOPS complex. Given that GFP-trap beads very efficiently deplete a lysate from all GFP tagged proteins, they should be able to detect an interaction with endogenous HOPS, even if only a fraction of TBC1D15 interacts with HOPS at any given point in time. This is really important as the binding to HOPS is the hypothetical recruitment mechanism through SKIP.

Minor points:

1. In the introduction, the authors cite Korolchuk, Saiki et al, 2011 for the discovery that lysosomes sense nutrients. That paper, while it is a nice one, has shown that lysosomal positioning influences nutrient sensing. The authors should cite one or two papers from Sabatini and colleagues, for example Sancak et al, 2008 in Science or Sancak et al., 2010 in Cell for this discovery.

2. Results section Figure 3: interphases should be interfaces

Referee #2:

In this manuscript, the authors have documented a cross-talk between two small GTPases located on late endosomes, Rab7 and Arl8b. This cross-talk is based on the concerted action of the Arl8b effector SKIP, the HOPS complex, and TBC1D15, a Rab7 GAP. They provide evidence that this process is involved in the dynamic distribution of late endosomes between the perinuclear region and the cell periphery.

Overall, the data are of high quality, and the model presented in Fig. 8 is attractive. However, it is not fully supported, and even seemingly contradictory to the results presented in the manuscript. My main concerns are the following:

- 1) What exactly is the role of SKIP? The authors provide evidence that SKIP interacts with Rab7. Is this interaction direct as suggested by the drawing shown in Fig. 3? If so, does SKIP binding to Rab7 interfere with RILP binding? In any case, the model does assume a direct interaction between Rab7 and SKIP.
- 2) It is proposed that TBC1D15 is targeted to SKIP-positive membranes through the HOPS complex. However, the respective roles of SKIP and the HOPS complex in the recruitment of TBC1D15 need to be clarified. Does depletion of SKIP prevent TBC1D15 recruitment?
- 3) An experiment showing that Arl8b, Rab7, SKIP, the HOPS complex and TBC1D15 (or at least the R416A mutant) can be co-immunoprecipitated in the same complex would greatly strengthen the model.

Referee #3:

The manuscript by Bakker et al. reports a switch from Rab7- to Arl8b- positive late endosomes/lysosomes mediated by the Rab7 GAP TBC1D15 and SKIP, a dual Arl8b and Rab7 effector. Overexpression of the Rab7 effector RILP, which associates with the minus-end-directed dynein/dynactin microtubule motor complex, was found to cause perinuclear accumulation of CD63-positive late endosomes/lysosomes as shown before. Interestingly, co-overexpression of SKIP resulted in a bimodal distribution of the CD63 endosomes/lysosomes, with one population being perinuclear and the other close to the plasma membrane. siRNA-mediated depletion of TBC1D15 caused a redistribution of a fraction of Rab7-positive endosomes/lysosomes towards the plasma membrane. On the other hand, siRNA-mediated depletion of the HOPS complex subunit Vps18 reduced the distribution of TBC1D15 close to the plasma membrane whereas overexpression of Vps18 had the opposite effect. SKIP-positive endosomes/lysosomes expressing Vps18, were found to be negative for Rab7. The authors propose a model whereby TBC1D15 is recruited to SKIP-positive membranes via the HOPS complex, thereby inactivating Rab7 on the same membranes (Fig 8).

This is a timely and potentially interesting manuscript, since the relationship between Arl8b and Rab7 on endosomes/lysosomes has so far remained poorly understood. However, a couple of issues need to be addressed prior to eventual publication.

Major points:

1. The characterization of SKIP as a Rab7 effector is central to this manuscript and needs to be performed more thoroughly. Direct association of SKIP with GDP- vs GTP-loaded Rab7 should be investigated *in vitro*, and the Rab7-binding domain of SKIP should be more closely defined.
2. The reported switch from Rab7- to Arl8b-positive endosomes/lysosomes is based on studies of fixed cells, but it would seem important to visualize it directly in live cells using fluorescently tagged Rab7 and Arl8b constructs in unperturbed cells as well as under conditions of TBC1D15/HOPS manipulation.

Minor point:

The authors should discuss their thoughts on why the cells "need" a Rab7- to Arl8b switch as long as Rab7-positive endosomes/lysosomes can also be transported in the plus end direction via the Rab7 effector FYCO1.

Dear Editors and Referees,

Below you can find the point-by-point responses to Reviewers' comments on our manuscript entitled 'SKIP-HOPS recruits TBC1D15 for a Rab7-to-Arl8b identity switch to control late endosome transport' (EMBOJ-2019-102301). We were greatly encouraged by the positive responses of the three Reviewers, all of whom found our study relevant and of high quality. Still, a number of important questions were raised to help substantiate our proposed model on the inactivation and release of Rab7 by the Arl8b/SKIP-associated transport complex. Briefly, the comments converged onto three main points, as follows:

Point 1: Demonstrate presence of Rab7 GAP TBC1D15 on endosomes under native conditions and distinguish the role of TBC1D15 on SKIP-positive endosomes from its previously established functions on mitochondrial TBC1D15, as well as from those of other known Rab7 GAPs TBC1D2 and TBC1D5 (requested by Reviewer 1).

Point 2: Test whether SKIP can bind Rab7 directly and solidify SKIP's role as an effector of Rab7 (requested by Reviewers 2 and 3).

Point 3: Strengthen the relationship between the HOPS complex and recruitment of TBC1D15 to SKIP (brought up by all Reviewers).

To address these points, we performed an array of interaction studies, as well as expanded the scope covered by imaging experiments in both fixed and live cells. Numerous new experiments have now been added to revised manuscript, and of the 10 main figures, only figure 3 remains unaltered since the first submission. The new data presented in the revised manuscript further support our initial findings and substantiate the proposed mechanism on the Rab7-to-Arl8b membrane identity switch coordinated by their shared effector SKIP. We thank the Reviewers for the interest expressed in our study and constructive criticisms offered to improve it. We hope the Editor and the Reviewers deem the revised manuscript ready for publication in the EMBO Journal.

Referee #1:

In the present manuscript, Bakker et al. first investigate the spatial and dynamic relationship between the small GTPases RAB7 and ARL8b. These GTPases mediate the directional movement of late endosomes/lysosomes in an opposing fashion through engagement of their effectors RILP and SKIP, respectively. The balance between RAB7-RILP and ARL8b and SKIP activity this determines late endosome localization and also determines their functionality. In their overexpression system, ARL8b and SKIP drive LE/Lys towards the cell periphery, concomitant with a removal of RAB7 from these structures. They present evidence that the RAB7 GAP TBC1D15 can shuttle from mitochondria to peripheral late endosomes through the action of SKIP and the HOPS complex, which removes RAB7 from these endosomes thereby promoting correct LE/ly mobility. Thus, TBC1D15 regulates RAB7 activity, late endosome positioning and also functionality.

The data is of high quality and largely supportive of the authors' hypothesis.

However, I do have serious concerns about the validity of the data as very important data are missing in the manuscript:

I. As far as I can tell, all the data on localization of GFP-TBC1D15 (endogenously tagged) has been obtained from cells overexpressing SKIP and or ARL8b. The authors need to obtain evidence that TBC1D15 is present on endosomal and or lysosomal membranes in parental cells which do not overexpress anything to support their thesis. TBC1D15 strongly localizes to mitochondria and at least some of the data that the authors present (loss of lysosomal motility, for example) could also be due to its role in the recently discovered lysosome-mitochondria contacts. Depletion of TBC1D15 should strengthen these contacts so that enhanced lysosome-mitochondria tethering then perturbs lysosomal mobility and positioning. The TBC1D15 antibody from the human proteome atlas (HPA013388, probably the same antibody as the one used in this study) works very well to detect endogenous TBC1D15 in fixed cells. If TBC1D15 does indeed play a role on lysosomes, the authors should be able to detect it there. GFP-TBC1D15 should also localize to lysosomes under native conditions. Depletion of SKIP or HOPS should then result in lack of TBC1D15 recruitment to lysosomes.

We thank the Reviewer for the complimentary assessment of the data presented in our first submission and the enthusiasm expressed towards our study.

We agree that ascribing localization of TBC1D15 to endosomes is important when considering this GAP's function in the Rab7-to-Arl8b transition. As requested by the Reviewer, we now show that endogenous non-mitochondrial GFP-TBC1D15 (visibly negative for TOM20 by immunofluorescence) colocalizes with endosomes labeled for endogenous Rab7 and CD63 (Fig 8A and B) in unperturbed HeLa cells. The same is also observed in cells depleted of FIS1 (Fig 8A and B), indicating that, at least in some instances, TBC1D15 is able to encounter its endosomal substrate without the need for (direct) mitochondrial involvement.

On Reviewer's suggestion, we also examined SKIP- and VPS18-depleted cells for localization of TBC1D15. However, due to the fact that mitochondria-independent localization of TBC1D15 to endosomes is an infrequent event, quantitative comparison between samples proved difficult. Further complicating matters, there was no reliable way to simultaneously assess depletion of target proteins in the same cells, as available antibodies were in conflict with other stainings in the sample.

While conducting these experiments we also realized that because the HOPS complex is a common denominator of several Rab7/effector complexes, it is possible that recruitment of TBC1D15 to endosomes via the HOPS complex could also occur in a SKIP-independent setting. It is important to note, however, that this consideration does not negatively impact our proposed model defining how SKIP/HOPS can influence Rab7 membrane occupancy through TBC1D15.

We thank the Reviewer for stimulating us to further explore mitochondria-independent function of TBC1D15 on endosomes. We feel that the new data featured in Fig 8, taken together with an array of additional FIS1 depletion

experiments described in detail under point #3 below, make an important addition to our study.

As it is, it is possible that overexpression of SKIP induces artificial recruitment of TBC1D15 to lysosomes. This is particularly relevant since HOPS components (VPS39) have been detected in lysosome-mitochondria contacts in yeast. Conceivably, SKIP/HOPS and TBC1D15 organize lysosome-mitochondria contacts and overexpression of SKIP results in a mislocalization of this machinery to lysosomes. The authors should also knock down FIS1, which will disrupt the mitochondrial localization of TBC1D15 and analyze RAB7 activity, lysosome positioning and the RAB7/ARL8b relationship. This could potentially serve to separately analyze the role of TBC1D15 on mitochondria and on lysosomes.

We have now included analysis of FIS1 depletion throughout the manuscript when discussing the function of TBC1D15. In none of the instances tested was loss of FIS1, judged highly efficient based on qPCR analysis (Fig 8F) and relocalization of TBC1D15 into the cytoplasm (Fig EV5A), able to recapitulate the effect(s) of TBC1D15 insufficiency (for details please see discussion under point #3 below). Together with the data on endosomal localization of TBC1D15, these results strongly support the notion that SKIP recruits non-mitochondrial TBC1D15 (or TBC1D15 released from mitochondria), and that this recruitment is not a result of artifactual diversion of machinery assembled by FIS1 on mitochondrial membranes for another function. We thank the Reviewer for the excellent suggestion to examine FIS1 in our study!

II. Also, the data on the TBC1D15-HOPS interaction is too preliminary for a quality journal like EMBOJ. The interaction is a core element of the authors' thesis and needs to be demonstrated more thoroughly (also see major points below).

This suggestion is addressed below under point #6.

III. Lastly, I was a bit surprised that the authors only looked at TBC1D15 and ignored the other two RAB7 GAP proteins (also see below).

To remedy this issue, we compared the effects of all 3 known Rab7 GAPs by siRNA-mediated silencing. The results, which are discussed in detail under point #2 below, fall nicely in line with our initial focus on TBC1D15 as the GAP mediating the choice between Arl8b and Rab7. In addition, the new data also offer an interesting twist on a possible competitive relationship between TBC1D15- and TBC1D5-promoted pathways emanating from the Rab7 endolysosome. We thank the Reviewer for prompting us to consider this.

In the major points listed below, I have summarized these and other concerns. Overall, the study is interesting and I am supportive of publication if (and only if) the authors can convincingly demonstrate the presence of TBC1D15 on endosomal membranes in the absence of overexpressed interaction partners.

Major points:

1. The authors should perform rescue experiments with TBC1D15. I realize that they have used multiple siRNAs but in the age of CRISPR/Cas9 they should also quickly knock out TBC1D15 and re-express it at low levels (lentiviral, preferably) to revert the resultant phenotype. They could also use a siRNA resistant cDNA. Maybe the authors tried this already and had problems with mis-localized, overexpressed TBC1D15? This should be stated as it would explain the somewhat outdated approach to use so many siRNAs.

To address this, we performed rescue of TBC1D15 deletion using siRNA oligo #3, which gave rise to a robust late compartment dispersion phenotype and strong retention of endogenous Rab7 on the SKIP compartment (Figs 7A and B and EV2A-C). Re-expression of GFP-TBC1D15, resistant to siRNA #3, nicely reinstated the organization of the perinuclear CD63-positive compartment following TBC1D15 silencing (Fig 6D and E). Unlike TBC1D15, expression of GFP-TBC1D5 in cells depleted with siTBC1D15 #3 could not alleviate endosomal dispersion (Fig 6D and E), indicating that the functions of these two Rab7 GAP proteins are not interchangeable with respect to the organization of late compartments.

2. The authors speculate that the control of RAB7 activity is required for efficient segregation of SKIP and RILP compartments. They then test the effect of TBC1D15 depletion. Given that TBC1D15 was reported to be localized to mitochondria, whereas two other RAB7 GAP proteins were reported to act on late endosomes, this appears to be a somewhat odd initial choice? Several independent reports have shown that TBC1D2/Armus regulates RAB7 activity on endosomes. Jimenez-Orgaz et al, 2018, EMBOJ and Seaman et al., 2018, JCS have recently shown that TBC1D5 acts through retromer and controls RAB7 activity in a major way. How are these mechanisms related? I would think that depletion of TBC1D2 and TBC1D5 should result in similar effects on RAB7 and ARL8b compartments? According to Jimenez-Orgaz et al (2018), and Seaman et al. (2018), RAB7 becomes grossly hyperactivated on LE/Lys in the absence of retromer/TBC1D5. Is the SKIP/HOPS/TBC1D15 mechanism simply a parallel pathway to retromer/TBC1D5? The authors should also cite these two studies in their discussion and maybe discuss whether these are related /redundant/parallel to their newly discussed pathway.

To address these questions (also related to the general point III above), we tested the effect of the 3 known Rab7 GAPs on the organization of the endolysosomal system. We found that loss of TBC1D2 (Armus) produced no appreciable effect on the intracellular distribution of late compartments; on the other hand, depletion of either TBC1D5 or TBC1D15 perturbed the normal arrangement of late endosomes and lysosomes (Fig 6A-C). Importantly, however, the nature of these alterations was markedly different between the latter two conditions. Specifically, knockdown of TBC1D5 appeared to induce a phenotype akin to overexpression of SKIP, causing accumulation of late endosomes at cell tips (Fig 6A-C). This suggested that, rather than functioning in the same pathway as SKIP, TBC1D5 instead

regulates a competitive pathway benefitting from removal of Rab7 from the same organelles (i.e. retromer-dependent retrieval).

By contrast, depletion of TBC1D15 resulted in intermediate localization of late compartments (Fig 6A-C), as expected in a tug-of-war scenario, where, instead of a regulated choice between transport routes, opposing motor complexes engaged on the same membranes (such as via Rab7 and Arl8b) are left to 'fight it out'.

Based on these observations, TBC1D15 appears to be the best candidate of the 3 known Rab7 GAPs to facilitate the choice between opposing late endosome transport routes. In support of this, as discussed under point 1 above, loss of TBC1D15 could only be rescued by re-expression of TBC1D15, but not TBC1D5 (Fig 6D, E), indicating that TBC1D15 fulfills a unique function in late compartment biology.

These considerations have now been added to the Results and Discussion sections of the manuscript, along with the suggested citations. We thank the Reviewer for bringing up this interesting issue, exploration of which both solidified our initial findings and expanded the scope of the study with regards to the existing literature.

3. Mitochondrial TBC1D15: As reported previously, the authors find endogenous TBC1D15 on mitochondria. What happens to SKIP/ARL8b or RAB7 when FIS1, the mitochondrial tether for TBC1D15, is depleted? FIS1 knockdown/KO results in a cytosolic dispersal of TBC1D15, which obviously greatly increases the free pool of TBC1D15. If the authors' thesis is correct, this should lead to changes in late endosome positioning. They could also use FIS1 depletion as a tool to separately analyze the function of TBC1D15 on lysosomes and on mitochondria.

As requested by the Reviewer (and related to the general point I above), we tested FIS1 depletion in a number of salient experiments from which key conclusions regarding the function of TBC1D15 were deduced. The observations in all cases were unequivocal: FIS1 does not appear to be involved in the segregation of late compartments as instigated by the endolysosomal effectors RILP and SKIP. Specifically, unlike loss of TBC1D15, knockdown of FIS1 does not result in dispersion of late compartments (Fig 8C, D and F), nor does it lead to retention of endogenous Rab7 on SKIP-positive membranes in the presence of RILP (Figs 8E and EV3C). Furthermore, acquisition of endogenous TBC1D15 by SKIP/HOPS is not affected by loss of FIS1 (Fig 9G and EV5A). Finally, colocalization of non-mitochondrial TBC1D15 with Rab7- and CD63-positive structures is still observed in FIS1-depleted cells (Fig 8A and B). In our view, these findings clearly support the notion that TBC1D15 cooperates with SKIP in a manner independent of FIS1.

4. Figure 4A: The authors use FRAP to investigate RAB7 membrane cycling as the only (indirect) tool to analyze the activity status/mobility of RAB7. Ideally, the authors would determine the nucleotide status of RAB7 directly. However, I realize this is not necessarily trivial so the authors could also use established effector assays such as the assay based on GST-RILP developed by Bucci and colleagues (PMID:19347311). Given that depletion of TBC1D15 results in a striking loss of

RAB7 mobility, indicative of an arrest in the active state, there should be much more activated RAB7 binding to the GST-RILP probe? How does the effect compare to loss of TBC1D2 and TBC1D5?

As stated by the Reviewer, probing the nucleotide status of Rab7 directly is challenging, and considering that we now examine the involvement of 3 GAP proteins previously described to target GTPase function of Rab7, it is unclear what such determination would add to the conclusions of the study. Accumulation of the GTP-bound Rab7 state under conditions of TBC1D15 inactivation has already been previously demonstrated by Peralta et al 2010, and a citation referring to this study now accompanies the FRAP experiment in the Results section of the revised manuscript.

5. Figure 1: The data on RILP and SKIP overexpression is quite clear. Yet, the authors could easily strengthen their conclusions by knocking down (or knockouts) endogenous SKIP and RILP. This, if the thesis is accurate, should lead to the opposite effects compared to overexpression.

These experiments were performed as requested and are now presented in Fig 5A-C of the revised manuscript. To investigate the effects of effector depletion in a fully endogenous setting, we created a HeLa cell line wherein the endogenous Arl8b was tagged with GFP (G-eArl8b). These cells were then depleted of either RILP or SKIP using siRNA pools and subsequently immunostained against Rab7. Depletion of either of the effectors led to an increase in colocalization between Rab7 and Arl8b (Fig 5A and B)—providing important contrast to sharply diminished Rab7:Arl8b overlap upon dual effector co-expression (Fig 2A and D). These data support our proposed model, wherein Rab7/RILP and Arl8b/SKIP mediate spatial compartmentalization of late organelles.

We appreciate the suggestion to include these data.

6. Figure 7B: I do not like that both tested interactors are overexpressed, I have seen far too many unspecific results when large amounts of two proteins are overexpressed in HEK293 cells. Given that an interaction between TBC1D15 and HOPS is a crucial part of their story, the presented IP data is too preliminary. The authors should test whether the RFP-tagged HOPS components precipitate endogenous TBC1D15. In addition, they also need to show that their endogenously GFP-tagged TBC1D15 precipitates the endogenous HOPS complex. Given that GFP-trap beads very efficiently deplete a lysate from all GFP tagged proteins, they should be able to detect an interaction with endogenous HOPS, even if only a fraction of TBC1D15 interacts with HOPS at any given point in time. This is really important as the binding to HOPS is the hypothetical recruitment mechanism through SKIP.

To probe deeper the involvement of the HOPS complex in the recruitment of TBC1D15 to SKIP, we undertook a series of approaches detailed below (also pertinent to the general point II above).

Already in the initial submission of the manuscript, we complemented the data for the interaction between TBC1D15 and VPS18 (referred to above) with imaging studies showing that depletion of VPS18 is detrimental to the acquisition of endogenous TBC1D15 by the SKIP compartment (Fig 10D and E), while, conversely, over-expression of VPS18 is beneficial in this regard (Fig 9G). These data imply that the interaction inferred from dual over-expression co-IP is biologically relevant.

On the suggestion of the Reviewer to probe the endogenous complex, we attempted co-precipitation of endogenous HOPS complex members with TBC1D15 but could not detect a stable interaction. This in and of itself is not surprising, considering that transient association of the GAP with the SKIP transport complex is likely beneficial for a smooth GTPase hand-over, thus preventing unwarranted or premature release of Rab7 from endosomal membranes.

To obtain additional evidence of HOPS involvement, we sought another way to biochemically access (transient) acquisition of TBC1D15 by SKIP. For this, we turned to proximity-based biotin ligation (BioID). The advantage of this technique over co-IP is that it allows detection of complexes formed in living cells, which may be destabilized/disrupted during lysis. To do this, we made use of a SKIP construct harboring a promiscuous biotin ligase domain at its N-terminus. In the presence of exogenously supplied biotin, biotinylation of GFP-TBC1D15 by BioID-SKIP was observed well above unspecific labeling afforded by free BioID moiety (empty vector control) under the same experimental conditions, despite the fact that the latter expressed at substantially higher levels (Fig 10A and B). In the same experiment, labeling of endogenous VPS18 by BioID-SKIP (above control) was also observed, indicating that both VPS18 and TBC1D15 are in complex with SKIP.

We then used this setup to evaluate the impact of VPS18 on complex formation between SKIP and TBC1D15. Partial depletion of VPS18 (reducing protein levels to 40-50% of control) in two different cell lines (HEK293T and HeLa) resulted in a measurable reduction of TBC1D15 biotinylation by BioID-SKIP (Fig 10F-H)—an observation echoing diminished recruitment of TBC1D15 to the SKIP compartment. Although biochemical interrogation of endogenous TBC1D15 was not successful in this context (insufficient labeling observed), we would like to stress that the results obtained with overexpressed GFP-TBC1D15 are fully consistent with all microscopy studies featuring its endogenous counterpart.

Taken together with the evidence provided in the first submission, the new experiments described above strongly support a model wherein the HOPS complex promotes recruitment of TBC1D15 to the SKIP compartment. We appreciate the Reviewer's constructive criticism and hope that the efforts put towards addressing it illustrate the seriousness with which we approached the issue.

Minor points:

1. In the introduction, the authors cite Korolchuk, Saiki et al, 2011 for the discovery

that lysosomes sense nutrients. That paper, while it is a nice one, has shown that lysosomal positioning influences nutrient sensing. The authors should cite one or two papers from Sabatini and colleagues, for example Sancak et al, 2008 in Science or Sancak et al., 2010 in Cell for this discovery.

These citations have been incorporated as requested. We appreciate the suggestion.

2. Results section Figure 3: interphases should be interfaces

This has been corrected. Our apologies for the error.

Referee #2:

In this manuscript, the authors have documented a cross-talk between two small GTPases located on late endosomes, Rab7 and Arl8b. This cross-talk is based on the concerted action of the Arl8b effector SKIP, the HOPS complex, and TBC1D15, a Rab7 GAP. They provide evidence that this process is involved in the dynamic distribution of late endosomes between the perinuclear region and the cell periphery.

Overall, the data are of high quality, and the model presented in Fig. 8 is attractive. However, it is not fully supported, and even seemingly contradictory to the results presented in the manuscript.

We thank the Reviewer for the complimentary assessment of the data and model presented in our study.

My main concerns are the following:

1) What exactly is the role of SKIP?

We propose that the role of SKIP is to target Rab7+/Arl8b+ hybrid identity multivesicular bodies for plus-end-directed transport to the cell periphery. To achieve this, SKIP acts as a negative effector of Rab7, as discussed below.

The authors provide evidence that SKIP interacts with Rab7. Is this interaction direct as suggested by the drawing shown in Fig. 3? If so, does SKIP binding to Rab7 interfere with RILP binding? In any case, the model does assume a direct interaction between Rab7 and SKIP.

Yes, the interaction with Rab7 is direct and mediated by the C-terminal portion of SKIP—in contrast to the N-terminal RUN domain needed to bind Arl8b. To arrive at this conclusion, a number of complementary experiments were performed, as follows.

In the first submission we demonstrated that SKIP interacts with Rab7 in a GTPase activity-dependent manner (Fig 4B, which are now quantified in 4C), suggesting that SKIP may function as an effector of Rab7. To explore this, a number of biochemical experiments were performed. We now show that full length SKIP can be successfully precipitated by recombinant GST-Rab7 (Fig 4D), indicating that the binding is likely direct. Notably, the apparent affinity of SKIP for GST-Rab7 is visibly less than that for GST-Arl8b, while the quality of the GST-Rab7 prep validates nicely against RFP-RILP in the same experiment (Figs 4D and EVIF), demonstrating robust binding. These results are consistent with the notion that the function of SKIP with respect to Rab7 is that of a negative effector (i.e. mediating disengagement of this GTPase from target membranes) and is thus by definition transient.

We went on to identify sequence determinants in SKIP mediating the interaction with Rab7. The Rab7 binding interface maps to the C-terminal segment of SKIP, as evidenced from precipitation of a panel of Myc-SKIP truncation mutants with recombinant GST-Rab7 (Fig 4E), as well as conventional co-IP experiments (Fig EVIH). The results demonstrate that SKIP utilizes an extensive interface to contact Rab7, as only the fragment spanning amino acids 537-1019 of SKIP supports full binding, while removal of either the C-terminal 874-1019 segment (resulting a construct spanning aa 537-873) or the middle segment 537-744 (resulting in a construct spanning aa 745-1019) is detrimental to the interaction. On this basis we conclude that recognition of Rab7 by SKIP involves sequence determinants different from those occupied by Arl8b.

To cement the status of SKIP as a bona fide Rab7 effector, we show that SKIP possesses a KMI variant (Fig 4E) of the canonical Rab7-interacting KML motif present in other effectors of this GTPase. We show that mutation of KMI residues (aa 610-612) to AAA in either full length SKIP or its C-terminal fragment 537-1019 (necessary and sufficient to attract Rab7) reduces Rab7 binding by half (Figs 4F, G and EVII), while the AAI mutant only affects the interaction by roughly 25%, indicating that 'I' contributes to Rab7 binding. Partial reduction in binding by AAA relative to wild type is further consistent with the truncation analysis indicating that the C-terminal 874-1019 SKIP segment is also a key contributor to the interaction with Rab7.

Taken together with the experiments provided in the first submission, the data discussed above demonstrate that SKIP follows the Rab7 effector paradigm of engagement with this GTPase but does so less efficiently likely to allow transient rather than persistent regulation. We appreciate the Reviewer's suggestion to develop this line of inquiry, which led us to describe SKIP as a new (negative) effector of Rab7.

2) It is proposed that TBC1D15 is targeted to SKIP-positive membranes through the HOPS complex. However, the respective roles of SKIP and the HOPS complex in the recruitment of TBC1D15 need to be clarify.

Our model posits that Arl8b-associated SKIP juxtaposes Rab7 with its GAP TBC1D15. In the initial submission we provided evidence that TBC1D15 is

recruited to SKIP-positive membranes by the HOPS complex, resulting in select inactivation and removal of Rab7. The role of SKIP is thus to bind both Rab7 and the HOPS complex, while the role of HOPS is to recruit TBC1D15. To substantiate the proposed mechanism, we dissected how SKIP organizes this Rab7 inactivation complex. Our new data reveal that the HOPS subunit VPS39, which has been described as a direct interactor of SKIP, helps to 'sense' Rab7 engagement. A detailed summary of these and other relevant findings accrued during the revision process appears below.

As described in detail under point 1) above, we have now identified a Rab7-interacting KMI motif in SKIP. We then used this information to examine whether binding Rab7 affects assembly of the HOPS complex onto SKIP. It has been shown previously that flanking HOPS subunits VPS39 and VPS41 make direct contacts to respectively engage SKIP and Arl8b (Khatler et al, 2015). We now find that the mutant SKIP lacking the KMI motif (SKIP-AAA) is unable to bind VPS39, as assayed by co-IP (Fig 9A and B), and thus fails to recruit this subunit into the HOPS complex (Fig 9C and D). At the same time, SKIP-AAA exhibits enhanced engagement with VPS41, the direct partner of Arl8b (Figs 9A-D and EV4C), and to some extent with the core HOPS subunits, whose binding to wild type SKIP is rather weak. These results imply that assembly of the full HOPS complex onto SKIP depends on direct contacts with Rab7 and suggest that SKIP/HOPS straddles Rab7 and Arl8b for a regulated identity switch.

The revised manuscript also provides an expanded understanding of TBC1D15 recruitment onto the SKIP/HOPS complex. In the first submission we showed that VPS18 and VPS16 interact with TBC1D15 (Fig 10C), and that loss of VPS18 inhibits TBC1D15 recruitment to the SKIP compartment (Fig 10D and E). Still, direct demonstration of TBC1D15 as a component of the SKIP complex was lacking. To remedy this, we employed a BioID approach for monitoring association of TBC1D15 with SKIP by fusing a promiscuous biotin ligase domain to SKIP. Using this technique, we now show that TBC1D15 comes in close proximity of SKIP (Fig 10A and B), and that loss of VPS18 negatively affects this interaction (Fig 10F-H). These findings (described in further detail under point 6 of Reviewer 1), biochemically corroborate VPS18-dependent association of TBC1D15 with SKIP and delineate a new function for the core of the HOPS complex in recruitment of a modulating factor for the Rab7 GTPase.

Does depletion of SKIP prevent TBC1D15 recruitment?

For discussion on this point please see the response to the general point I from Reviewer 1.

3) An experiment showing that Arl8b, Rab7, SKIP, the HOPS complex and TBC1D15 (or at least the R416A mutant) can be co-immunoprecipitated in the same complex would greatly strengthen the model.

We appreciate the Reviewer's interest in our proposed model and constructive criticism offered to substantiate and expand it. Needless to say, isolation of a likely

transient membrane-associated complex comprised of so many proteins is not a trivial task. Although we do not directly show that Arl8b, Rab7, SKIP, HOPS and TBC1D15 can all be precipitated in the same protein complex, a number of key pieces of strongly substantiating evidence for the existence of such a complex have been obtained during the revision. These include:

-inability of SKIP lacking the RUN domain to function on endosomal membranes (Fig 9C and D). This observation implies that all effects of SKIP on endosomal Rab7 occur in association with Arl8b, therefore substantiating the existence of the Arl8b/SKIP/Rab7 complex.

-requirement for direct binding to Rab7 on the part of SKIP (via the KMI motif) in order to recruit the full HOPS complex, as illustrated by lack of VPS39 association with the KMI-AAA mutant of SKIP (Fig 9A-D). By contrast, the other tested subunits of the HOPS complex can still be recruited to SKIP-AAA, presumably via Arl8b (Fig 9A-D). In fact, association of VPS41 (the direct partner of Arl8b), as well as that of the core subunits VPS11, 18, and 16, is enhanced by SKIP-AAA, suggesting that wild type SKIP 'shares' the HOPS with both GTPases. Taken together these findings substantiate the existence of a complex containing Arl8b/SKIP/HOPS/Rab7 and position VPS39 as a Rab7 'sensor', which is acquired by SKIP somewhat independently of the rest of the HOPS.

-ability of HOPS to modulate acquisition of TBC1D15 by SKIP (in both over-expression and depletion settings presented in Figs 9 and 10, respectively) substantiates the existence of the SKIP/HOPS/TBC1D15 complex, while ability of TBC1D15 to displace Rab7 from membranes occupied by SKIP/HOPS in an activity-dependent manner substantiates presence of Rab7 in the same complex.

We hope the Reviewer finds these multifaceted considerations helpful in supporting publication of our revised manuscript.

Referee #3:

The manuscript by Bakker et al. reports a switch from Rab7- to Arl8b- positive late endosomes/lysosomes mediated by the Rab7 GAP TBC1D15 and SKIP, a dual Arl8b and Rab7 effector. Overexpression of the Rab7 effector RILP, which associates with the minus-end-directed dynein/dynactin microtubule motor complex, was found to cause perinuclear accumulation of CD63-positive late endosomes/lysosomes as shown before. Interestingly, co-overexpression of SKIP resulted in a bimodal distribution of the CD63 endosomes/lysosomes, with one population being perinuclear and the other close to the plasma membrane. siRNA-mediated depletion of TBCD15 caused a redistribution of a fraction of Rab7-positive endosomes/lysosomes towards the plasma membrane. On the other hand, siRNA-mediated depletion of the HOPS complex subunit Vps18 reduced the distribution of TBCD15 close to the plasma membrane whereas overexpression of Vps18 had the opposite effect. SKIP-positive endosomes/lysosomes expressing Vps18, were found to be negative for Rab7. The authors propose a model whereby TBC1D15 is recruited to SKIP-positive membranes via the HOPS

complex, thereby inactivating Rab7 on the same membranes (Fig 8).

This is a timely and potentially interesting manuscript, since the relationship between Arl8b and Rab7 on endosomes/lysosomes has so far remained poorly understood. However, a couple of issues need to be addressed prior to eventual publication.

We thank the Reviewer for their positive evaluation of our manuscript and constructive criticisms offered to improve it. Specific points raised are addressed below.

Major points:

1. The characterization of SKIP as a Rab7 effector is central to this manuscript and needs to be performed more thoroughly. Direct association of SKIP with GDP- vs GTP-loaded Rab7 should be investigated *in vitro*, and the Rab7-binding domain of SKIP should be more closely defined.

*To delve deeper into the relationship between SKIP and Rab7 we performed *in vitro* binding assays and characterized sequence determinants in SKIP necessary to bind Rab7. The following new data have now been added to the revised manuscript:*

-Demonstration of SKIP binding to recombinant GST-Rab7 (evaluated in comparison to GST-Arl8b) (Fig 4D).

*-Characterization of the interaction interface with Rab7, mapping to the C-terminal segment of SKIP, as assayed both *in vitro* (Fig 4E) and by co-IP (Fig EV1H). These experiments demonstrate that Rab7 targets different sequence determinants of SKIP as compared to Arl8b, consistent with its proposed role in negotiating a regulated hand-over transition.*

*-Identification of a KMI variant of the canonical Rab7-interacting KML motif in SKIP (aa 610-612, Fig 4E), providing evidence of direct contacts with the Rab7 GTPase. We went on to show that KMI-to-AAA mutant of SKIP exhibits compromised binding to Rab7 (Fig 4F and G) and fails to associate with the VPS39 subunit of the HOPS complex (Fig 9A-D), previously shown to be a direct binding partner of SKIP (Khatter et al, 2015). Although no explicit biochemical distinction is made between GDP- and GTP-loaded states in the *in vitro* binding experiments, the co-IP data in Fig 4B clearly demonstrate SKIP's strong preference for constitutively active Rab7 Q67L over wild type, as well as a near complete loss of binding to the dominant negative variant T22N. These data are now quantified in Fig 4C from 3 independent experiments.*

For further details on the aforementioned results please see responses to point 1) of Reviewer #2 above. Taken together with the evidence provided in the first submission, these results establish SKIP as a bona fide effector of Rab7 tasked with a unique role of disengaging Rab7 from endosomal membranes. We thank the Reviewer for prompting us to further develop this aspect of our model.

2. The reported switch from Rab7- to Arl8b-positive endosomes/lysosomes is based on studies of fixed cells, but it would seem important to visualize it directly in live cells using fluorescently tagged Rab7 and Arl8b constructs in unperturbed cells as well as under conditions of TBC1D15/HOPS manipulation.

We agree with the Reviewer that observation of endosome behavior in real time is invaluable when seeking to understand the mechanisms governing these dynamic organelles. To address this, we performed two new experiments in live cells.

Firstly, the effects of Rab7 function on the organization and dynamics of the Arl8b compartment were explored, where we discovered that Rab7 activity status modulates localization and motility of vesicles labeled by Arl8b. Two observations stood out. Firstly, co-expression of constitutively active mCherry-Rab7 Q67L resulted in disorganized movement of Arl8b vesicles throughout the cell, as compared to the wild type mCherry-Rab7 condition (Fig 1C and D). Secondly, under co-expression of dominant negative mCherry-Rab7 T22N, the peripheral Arl8b repertoire was largely absent, and movement of remaining Arl8b-positive organelles (now relegated almost exclusively to the perinuclear region) was severely hampered (Fig 1C-E). These observations imply that Rab7 activity is crucial for Arl8b-mediated transport to commence.

We now use this experiment as an opening argument to set up the investigation of the interplay between Rab7 and Arl8b-associated endosomal machineries and thank the Reviewer for prompting us to expand our analysis of endosome behavior in live cells.

We further attempted to apply the above experimental setup (i.e. ectopic expression of Arl8b-GFP and mCherry-Rab7) to the TBC1D15 depletion setting. However, we found that overexpression following knockdowns made it difficult to follow the switch between a double positive compartment to Arl8b only in real time without the addition of the effectors, which quickly became too complex to achieve in the timeframe allotted for the revision.

We did however succeed with another approach to visualize the first part of the switch—selection of Rab7-positive endosomes by SKIP prior to the generation of the peripheral SKIP compartment at cell tips. To do this, we designed a SKIP construct whose targeting to membranes could be chemically induced (Fig 4I). Using this system, we show that SKIP selects Rab7-positive membranes in early steps of plus-end-directed transport of vesicles to the cell periphery, nicely complementing our studies on SKIP possessing canonical Rab7 effector determinants (Fig 4B-G).

We thank the Reviewer for stimulating us to pursue this avenue of investigation.

Minor point:

The authors should discuss their thoughts on why the cells "need" a Rab7- to Arl8b switch as long as Rab7-positive endosomes/lysosomes can also be transported in the plus end direction via the Rab7 effector FYCO1.

This is indeed a very interesting avenue to contemplate, especially considering the fact that the Rab7 binding motif of SKIP closely resembles that of FYCO1, more so in fact than that of PLEKHM1 (Fig 4E). We have now added a section to the Discussion elaborating on this issue (lines 430-445). We appreciate the invitation to delve into this comparison.

2nd Editorial Decision

18th Dec 19

Thank you for submitting a revised version of your manuscript. It has now been seen by the original referees, whose comments are shown below.

As you will see the reviewer find that all criticisms have been sufficiently addressed and recommend the study for publication. However, there are a few editorial issues concerning text and figures that I need you to address before we can draft a letter of official acceptance.

REFEREE REPORTS

Referee #1:

The authors have performed a significant amount of additional experimentation that largely addresses my previous concerns. Importantly, they have shown that TBC1D15 indeed resides on late endosomes in the absence of any manipulations.

I am now supportive of publication without further revisions.

Referee #2:

The authors have met my previous comments/criticisms in a very satisfactory way. The manuscript have been extensively revised and improved.

Referee #3:

The authors have successfully addressed the points I raised.

Corresponding Author Name: Jacques Neefjes and Ilana Berlin

Manuscript Number: EMBOJ-2019-102301